# RNA structure drives interaction with proteins

Natalia Sanchez de Groot [1,8], Alexandros Armaos[1,8], Ricardo Graña-Montes[1,7], Marion Alriquet[2,3], Giulia Calloni[2,3], R. Martin Vabulas[2,3] & Gian Gaetano Tartaglia[1,4,5,6]

The combination of high-throughput sequencing and in vivo crosslinking approaches leads to the progressive uncovering of the complex interdependence between cellular transcriptome and proteome. Yet, the molecular determinants governing interactions in protein-RNA networks are not well understood. Here we investigated the relationship between the structure of an RNA and its ability to interact with proteins. Analysing in silico, in vitro and in vivo experiments, we find that the amount of double-stranded regions in an RNA correlates with the number of protein contacts. This relationship —which we call structure-driven protein interactivity— allows classification of RNA types, plays a role in gene regulation and could have implications for the formation of phase-separated ribonucleoprotein assemblies. We validate our hypothesis by showing that a highly structured RNA can rearrange the composition of a protein aggregate. We report that the tendency of proteins to phase-separate is reduced by interactions with specific RNAs.

[1] Centre for Genomic Regulation (CRG), The Barcelona Institute for Science and Technology, Dr. Aiguader 88, 08003 Barcelona, Spain. [2] Buchmann Institute for Molecular Life Sciences, Goethe University Frankfurt, 60438 Frankfurt am Main, Germany. [3] Institute of Biophysical Chemistry, Goethe University Frankfurt, 60438 Frankfurt am Main, Germany. [4] ICREA 23 Passeig Lluis Companys 08010 and Universitat Pompeu Fabra (UPF), 08003 Barcelona, Spain. [5] Department of Biology 'Charles Darwin', Sapienza University of Rome, P.le A. Moro 5, Rome 00185, Italy. [6] Department of Neuroscience and Brain Technologies, Istituto Italiano di Tecnologia, Via Morego 30, 16163 Genoa, Italy. [7] Present address: Department of Biochemistry, University of Zürich, Winterthurerstrasse 190, 8057 Zürich, Switzerland. [8] These authors contributed equally: Natalia Sanchez de Groot, Alexandros Armaos. Correspondence and requests for materials should be addressed to R.M.V. (email: vabulas@em.uni-frankfurt.de) or to G.G.T. (email: gian.tartaglia@crg.eu)

Since the central dogma was proposed in 1950, the main role attributed to RNA has been to act as the intermediate between DNA and protein synthesis. Yet, more than 70% of the genome is transcribed and just a small part codes for proteins[1,2], which indicates that the majority of RNAs could have different biological roles. During the past decade many efforts were made to develop methods to study RNA isoforms: sequencing has been essential for detection of RNA species[3] and recent developments provided a great deal of data on polymorphisms[4], expression[5] and half-lives[6] of all types of RNAs, generating a valuable resource to understand their cellular functions and regulation. Although a number of techniques identified biological characteristics such as cellular location[7] and secondary structure[8,9], the characterization of the interaction network remains one of the most urgent challenges[10,11]. To this aim, computational methods are being developed to identify physicochemical features of the transcripts[10], their conservation between species[12] and, most importantly, binding partners[13] that are also active in the cellular environment[14].

RNA is involved in many cellular processes such as control of gene expression, catalysis of various substrates, scaffolding of complex assemblies, and molecular chaperoning[15]. Its ability to act as a hub of cellular networks is at the centre of an active research field and has already led to the discovery of diverse ribonucleoprotein (RNP) assemblies[16,17]. A number of membrane-less organelles contain specific mixtures of RNAs and RBPs (RNA-binding proteins) that, due to their intrinsic lability, are difficult to characterize[10]. In most cases, liquid-like RNP assemblies, or condensates, such as P-bodies and stress granules[18], exchange components with the surrounding content and adapt to the environmental condition in a dynamic way. Within these phase-separated assemblies RNA plays a central role[19]: whereas a polypeptide of 100 amino acids can interact with one or two proteins, a chain of 100 nucleotides is able to bind to 5–20 proteins, thus providing an ideal platform or scaffold for interactions[20,21]. Not surprisingly, changes in the interactions within RNP granules leading to liquid-to-solid phase transition are associated with the development of several human diseases, including neurological disorders and different types of cancer[17]. In RNP condensates such as stress granules, regulation of protein and RNA contacts is primarily controlled by HSP70 and cochaperones[17] that act as versatile elements promoting assembly and disassembly of complexes[22].

In this large spectrum of activities, RNA structure controls the precise binding of proteins by creating spatial patterns and alternative conformations for the interactions to occur[12]. Known complexes in which the structure regulates protein binding include transfer RNAs (tRNAs) whose three-dimensional conformation facilitates the codon/anticodon interaction[23] and the ribosomal RNA (rRNA) scaffold that sustains the ribosome[24]. Importantly, the structure of a messenger RNA (mRNA) defines its lifecycle[25], recruitment of ribosomes and response against environmental changes[25]. There are several cases of nucleotide chains of non-coding RNAs acting as scaffolds for protein complexes[21]: structured domains in *NEAT1* attract paraspeckle components[26] and repeat regions in *XIST* sequester proteins to orchestrate X-chromosome inactivation[27]. By contrast, poorly structured snoRNAs have been shown to facilitate the assembly of other transcripts[28].

Of both coding and non-coding transcripts, RBPs are known as the major regulators[29] and are classified as single-stranded RNA (ssRNA) and double-stranded RNA (dsRNA), depending on their binding preferences. Here we investigated the relationship between RNA structure and ability to interact with RBPs. At the transcriptome level, we find that the amount of RNA secondary structure correlates with the number of protein interactions. We propose several possible implications of this relationship: a link to RNA types and biological roles; a connection to regulatory networks; and the ability to modulate phase separation. Based on our observations, we also demonstrated that this RNA property can be exploited in vitro to tune the contact network of a protein aggregate.

## Results

**Highly structured RNAs bind a large amount of proteins.** With the aim of studying how RNA structure influences protein binding, we measured the amount of double-stranded regions of the human transcriptome[8] (Fig. 1a). We first grouped the RNAs, as detected by enhanced crosslinking and immunoprecipitation (eCLIP) approach[30], in classes based on the structural content measured by 'parallel analysis of RNA structure' (PARS)[8] (Supplementary Fig. 1a and Fig. 1b). PARS is an experimental technique that distinguishes double- and single-stranded regions of RNA using the catalytic activity of two enzymes, RNase V1 (able to cut double-stranded nucleotides) and S1 (able to cut single-stranded nucleotides) and for which positive scores indicate double-stranded regions (see Eq. (1) in Methods)[8]. We then used catRAPID predictions of protein–RNA interactions (available from the RNAct database that contains both proteome-wide and transcriptome-wide calculations[31]) and compared the interaction scores of different groups (HS, high structural content, vs. LS, low structural content) (Fig. 1b). The catRAPID algorithm[32] estimates the binding potential through van der Waals, hydrogen bonding and secondary structure propensities of both protein and RNA sequences (total of 10 properties), allowing identification of binding partners with high confidence. Indeed, as reported in a recent analysis of about half a million of experimentally validated interactions[31], the algorithm is able to separate interacting vs. non-interacting pairs with an area under the curve (AUC) receiver operating characteristic (ROC) curve of 0.78 (with false discovery rate (FDR) significantly below 0.25 when the Z-score values are >2). Comparison of RNA groups with different structural content shows a consistent trend in which higher structural content in RNA molecules results in higher protein interaction scores (Fig. 1b). As for the PARS data, we note that the amount of double-stranded regions correlates weakly (<0.10; Pearson's) with RNA length and GC content, indicating that these two factors positively contribute to the secondary structure by increasing the size of the conformational space as well as the overall stability[33].

We repeated the analysis with a unrelated approach, RPISeq, which predicts protein–RNA interactions using sequence patterns in nucleotide and amino acid sequences[11]. RPISeq is comprised of two methods based on support vector machines (RPISeq-SVM) and random forest (RPISeq-RF). Due to specific computational requirements, we applied RPISeq to an ensemble of RBPs (50 proteins with sequence similarity <0.85; http://cd-hit.org/) against the HS and LS set from the tails of the structural content distribution (100 transcripts) to estimate the binding probabilities (Supplementary Data 1). In both cases, the HS set (RF 0.80, SVM 0.71) is predicted to bind with significantly higher probabilities than the LS set (RF 0.70, SVM 0.54; *p* value <10⁻⁵; Kolmogorov–Smirnov (KS) test; Supplementary Fig. 1b–c), in agreement with catRAPID analysis (Fig. 1b). Thus, our analysis suggests that the RNA structure content has effect on the interaction with proteins.

To match our predictions with experimental data, we investigated all RBP–RNA interactions revealed by enhanced CrossLinking and ImmunoPrecipitation, eCLIP[30] (118 RBPs; see Methods). eCLIP provides protein contacts on target RNAs at individual nucleotide resolution through ligation of barcoded

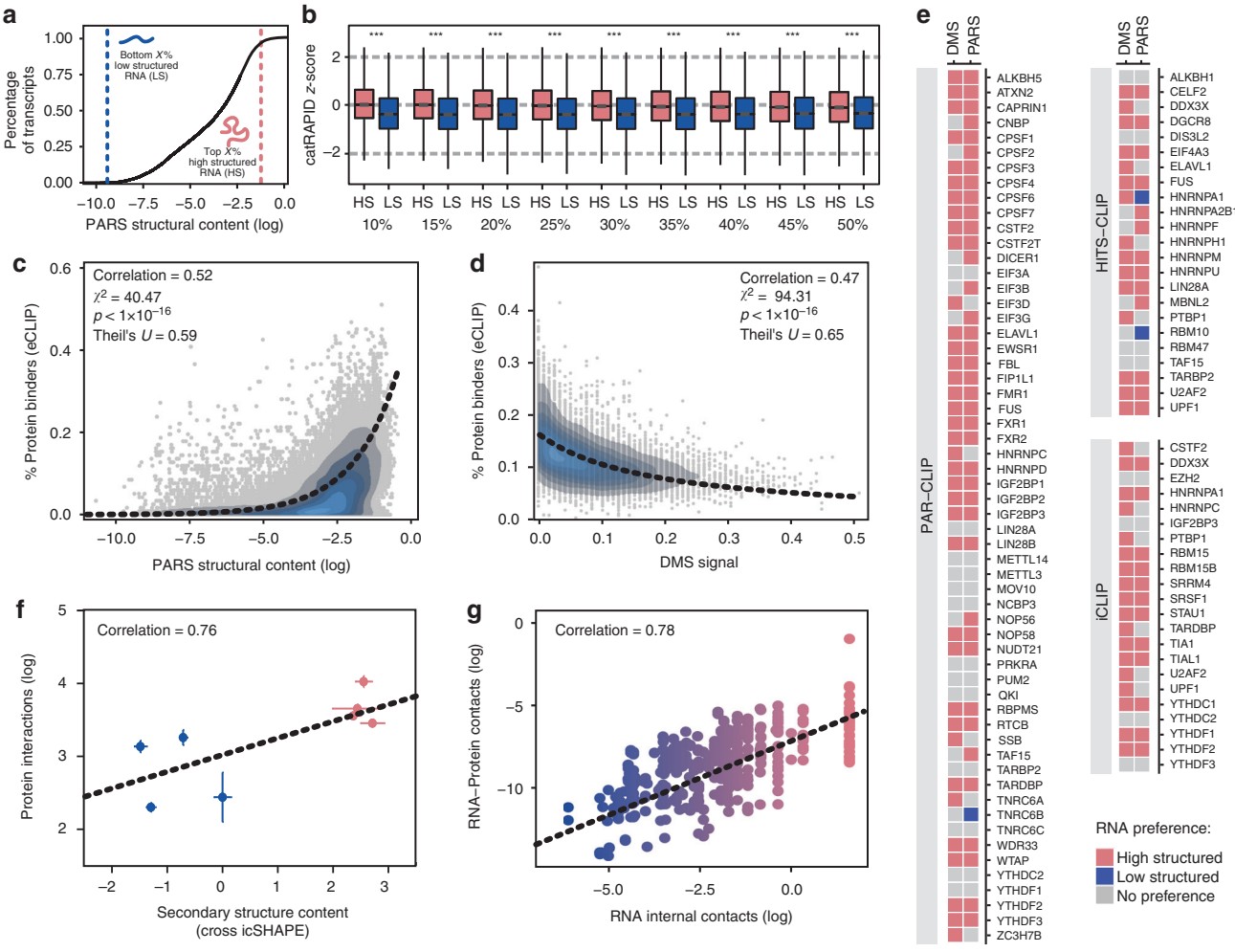

**Fig. 1** The amount of protein structure correlates with the number of interactions. **a** Cumulative distribution function (CDF) for the secondary structure content of all human RNAs measured by parallel analysis of RNA structure (PARS)[8,69]. Vertical lines indicate a certain fraction ($X$%) of RNAs with the lowest secondary content (LS; blue) and the same fraction with the highest secondary content (HS; pink). **b** catRAPID predictions of protein interactions with human RNAs ranked by structural content measured by PARS (118 RNA-binding proteins (RBPs) for which enhanced crosslinking and immunoprecipitation (eCLIP) information is also available)[31]. The fractions 10%, 15%, ..., 50% refer to the comparison between equal-size HS and LS sets. The results indicate that catRAPID is able to distinguish HS and LS groups significantly and consistently through the different fractions ($p$ value <$10^{-16}$; Kolmogorov–Smirnov (KS) test). The boxes show the interquartile range (IQR), the central line represents the median, the whiskers add 1.5 times the IQR to the 75 percentile (box upper limit) and subtract 1.5 times the IQR from the 25 percentile (box lower limit). s.d. is shown. **c** Relationship between number of protein interactions (eCLIP) and structural content measured by PARS[30]. The fitting line corresponds to the formula $y = \exp(\alpha + \beta x)$, where $\alpha = -0.75$; $\beta = 0.67$; $p$ value estimated with KS test. **d** Relationship between number of protein interactions and structural content measured by dimethyl sulfate modification (DMS)[9]. The fitting line corresponds to the formula $y = 1/(\alpha + \beta x)$, where $\alpha = 2.60$; $\beta = 87.36$; $p$ value estimated with KS test. **e** Structural preferences of RBPs measured with three different CLIP techniques (photoactivatable ribonucleoside-enhanced CLIP (PAR-CLIP), high-throughput sequencing-CLIP (HITS-CLIP) and individual nucleotide resolution CLIP (iCLIP)). The colour indicates the RNA-binding preference of each protein: pink, high structured; blue, low structured; grey, no preference. **f** Correlation between structural content (CROSS predictions of icSHAPE experiments) and protein interactions of eight transcripts revealed by protein microarrays (Pearson's correlation). s.d. is shown. **g** Analysis of Protein Data Bank (PDB) structures containing protein–RNA complexes reveals a trend between protein (inter) and RNA (intra) contacts (196 different pairs; Pearson's correlation)

single-stranded DNA adapters[30]. In agreement with catRAPID predictions[31] (Fig. 1b), eCLIP binding scores correlate with PARS secondary structure, which indicates that the RNA propensity to interact with proteins is proportional to the amount of structure measured transcriptome wide (Fig. 1c). We note that the CLIP-seq approaches in general favour detection of single-stranded (SS) RNA at the expense of double-stranded (DS) RNA[34] and the eCLIP dataset is not enriched in double-stranded RNA-binding proteins (9 out of 118 are assigned according to UniProt as dsRNA binding, 12 out of 118 as ssRNA binding, using available GO annotations[35]), which indicates that our results are not biased by the protein types used in our analysis.

To further corroborate that the trend is genuine and not only intrinsic to PARS measurements, we analysed the protein-interacting potential of the entire human transcriptome against the RNA secondary structure measured with the dimethyl sulfate modification (DMS) technique (differently from PARS, high values indicate single-stranded regions; Fig. 1d)[9]. This method of assessing RNA structure employs deep sequencing to detect unpaired adenosine and cytidine nucleotides. Once more, the analysis shows that the RNA secondary structure of the human transcripts is tightly correlated with protein-binding abilities.

We also used the POSTAR database (containing >1000 CLIP-seq datasets; http://lulab.life.tsinghua.edu.cn/postar/) to retrieve

the RNA-binding preferences of human proteins (103 experiments, 85 different RBPs) measured with PAR-CLIP, high-throughput sequencing-CLIP (HITS-CLIP) and individual nucleotide resolution CLIP (iCLIP)[10]. Due to intrinsic differences in the CLIP approaches (and other factors, such as the cell lines employed), each experiment reports different protein–RNA interactions[10]. Yet 77% of the RBPs have preference for highly structured RNAs for at least one of the experimental methods (DMS or PARS; Fig. 1e).

Given possible technical biases of high-throughput experiments, we decided to verify the reproducibility of the trend by investigating the correlation between RNA structure and protein interactions in low-throughput analyses. We first studied the interactome of eight large (>1000 nt) RNAs whose protein partners have been identified by microarray, a crosslinking-free approach[21,36,37] (see Methods). In parallel, we estimated the structural content of each transcript using the CROSS algorithm that was previously trained on SHAPE data[38] to predict the double-stranded propensity at nucleotide level resolution. Our results presented in Fig. 1f indicate that highly structured transcripts have more protein contacts than poorly structured transcripts, which is fully compatible with the findings presented in our previous analysis (Fig. 1b–e).

We corroborated our observations through the study of RNP complexes deposited in the Protein Data Bank (PDB) database (X-ray resolution <2 Å; Supplementary Data 2; see Methods), which is comprised of 196 distinct RNA–protein pairs (>20 species) analysed with different techniques (mainly X-ray and nuclear magnetic resonance (NMR)) by different laboratories. Measuring the amount of RNA intra-contact (i.e. amount of RNA structure) and inter-contact (i.e. amino acid) per nucleotide chain, we found a striking correlation of 0.78 between the two variables, which provides compelling evidence of their tight relation (Fig. 1g; see Eqs. (2) and (3) in Methods).

Thus, independently of the experiment (PARS, DMS, microarray, X-ray, NMR, eCLIP, PAR-CLIP, HITS-CLIP and iCLIP), the algorithms employed (catRAPID and RPISeq or CROSS to mimic SHAPE data) or organism (PDB database), we found a correlation between number of protein interactions and RNA structural content.

**The structure-driven protein interactivity of RNA types**. We next investigated if the tight link between secondary structure and number of protein interactions is a property of specific RNA types (Fig. 2a). To this aim, we compared the secondary structure and protein interactions of transcripts ranked by sequence similarity using the CD-HIT algorithm[39] (http://cd-hit.org/). With a threshold of 85% similarity, we found 22 clusters (total of 55 transcripts) with at least one RBP contact revealed by eCLIP. We then calculated the correlation between DMS signal and eCLIP protein interactions for each cluster and obtained a negative correlation in 64% of cases. This finding indicates that between two similar transcripts the one with higher structural content is more likely to have a larger number of protein interactions.

The two transcripts sharing the highest similarity (99.31%) are the γ-globins HBG1 and HBG2 (haemoglobin subunits γ1 and γ2) that are expressed in fetal liver, spleen and bone marrow (NCBI Gene ID: 3048). The γ-globin variant with higher structure (HBG1) has a significantly larger number of protein interactors (HBG1, average DMS signal of 0.04, 29 interactors; HBG2, average DMS signal of 0.07, 14 interactors; p value = 0.003; KS test; Fig. 2b). While the nucleotide composition of the two transcripts remains nearly the same (HBG1:280c, 463c, 514t, 552a, 575g; HBG2: 280t, 463g, 514g, Δ552a, 574a), the differences between HBG1 and HBG2 are concentrated in regions where the

secondary structure is altered (Supplementary Fig. 2). These results indicate that protein interactivity is tightly associated with conformational changes in elements of secondary structure. Interestingly, the increased double-stranded content in HBG1, especially in the 3′-UTR, is accompanied by an accumulation of translation regulatory elements (Fig. 2b) and a concomitant decrease in expression (NCBI Gene ID: 3048).

We then wondered whether specific RNA structures are involved in protein regulation. We divided the human transcriptome in different classes and analysed their secondary structure as detected by two independent experimental techniques, PARS and DMS. Both techniques show that protein-coding RNAs have the largest structural content (Fig. 2c, Supplementary Table 1)[38]. Although part of the mRNA structure is concentrated in the UTRs[8], when these are excluded, the distribution of the structural content does not change substantially (Pearson's correlation between transcripts with and without their UTRs = 0.94; Supplementary Fig. 3). The RNAs known to interact with proteins, such as small nuclear RNA (snRNAs)[40] and small nucleolar RNAs (snoRNAs)[28], show the highest amount of structure, whereas RNAs targeting complementary regions in nucleic acids such as antisense, miRNAs and a number of long intergenic non-coding RNAs (lincRNAs)[41,42] feature the smallest amount of structure[43] (Supplementary Table 1).

In agreement with our findings, Seemann et al.[12] previously observed a tight relationship between protein binding and conservation of structural elements in mRNAs, which occur to a lesser extent in long non-coding RNAs[12]. Although lincRNAs show a lower amount of double-stranded regions (lowest in PARS, third-lowest in DMS), we note that some of them, such as for instance NEAT1[44] and XIST[27], are able to scaffold protein assembly through structured domains. As there is an ongoing debate on the structural differences between coding and non-coding transcripts[45,46] and our analysis of DMS and PARS data reveals contradicting results for specific RNA types, we suggest further investigations in future studies (Fig. 2c; Supplementary Table 1).

To investigate functional differences between highly and poorly structured RNAs, we analysed GO terms associated to the least and most structured RNAs (100 LS vs. 100 HS transcripts) using the cleverGO[35] approach. While the LS set (14 non-coding RNAs and 86 mRNAs) is not associated with specific semantic similarity clusters (total of 36 terms with p value <0.05; Bonferroni test), the HS set (100 mRNAs; total of 395 terms with p value <0.05 and 103 terms with p value <0.01; Bonferroni test; Fig. 2d) includes 20 distinct clusters. The five main categories associated with the clusters and covering at least a quarter of the entries are: (i) complex protein regulation (49/103), (ii) nucleoside metabolic process (39/103), (iii) cellular response (29/103), (iv) gene expression (29/103) and (v) protein targeting (28/103). We also repeated the GO term analysis using as a background the 25% higher expressed transcripts and obtained similar results (K562 strain GENCODE, Methods, Supplementary Fig. 4).

The cluster analysis reveals the intriguing finding that transcripts with strong structural content interact more with polypeptides and code for proteins involved in regulatory functions and in the formation of complex contact networks. Given the relationship between RNA structure and number of protein interactions (Fig. 1), one preliminary interpretation of our results is that a high degree of control is required for genes that coordinate the activity of a large number of cellular networks[47]. Thus, our analysis suggests a 'recursive' property: highly contacted transcripts code for highly contacting proteins (Fig. 2e)[20,48].

**Disorder and helix distinguish dsRNA vs. ssRNA**. To understand the molecular basis of the structure-driven interactivity of

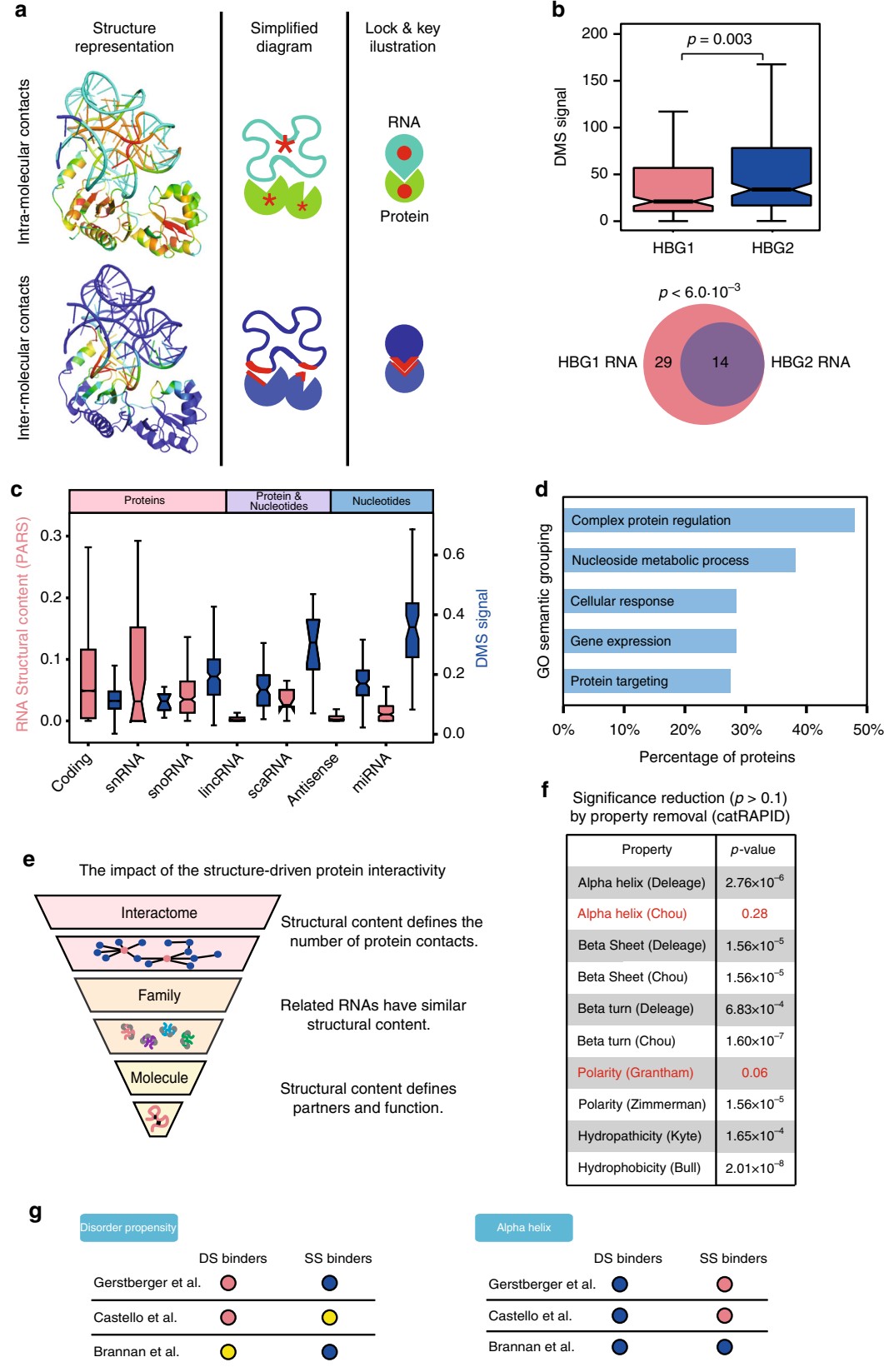

RNA molecules, we analysed which physicochemical properties of the proteins better discriminate the HS and LS sets. We studied all 10 variables used in the catRAPID algorithm (Fig. 2f)[13,32] and removed them one by one to estimate the impact on the prediction of RNA–protein interactions. We found that the capacity

to distinguish between the least and most structured RNAs (100 HS and LS transcripts; Supplementary Data 3) sets is more affected when the polarity ($p$ value = 0.28; KS test) and α-helical propensity ($p$ value = 0.06; KS test) are removed (Fig. 2f). The property that more significantly affects the HS binding propensity

**Fig. 2** Functional footprints of the RNA structure-driven protein interactivity. **a** Scheme showing the role of intra- and intermolecular contacts in a RNA–protein complex. Top, intramolecular contacts. Bottom, inter-molecular contacts. The number of contacts range is indicated with shades from dark blue (lowest) to red (highest). **b** Up, Structural content (dimethyl sulfate modification (DMS); $p$ value estimated with KS test). Bottom, Protein interactions (enhanced CrossLinking and ImmunoPrecipitation (eCLIP) of haemoglobin subunit γ1 (HBG1) (pink) and haemoglobin subunit γ2 (HBG2) (blue) RNAs (99.3% of sequential identity); the empirical $p$ value was estimated by comparing the overlap with that of 1000 samples taken from eCLIP RNA-binding proteins (RBPs). **c** Parallel analysis of RNA structure (PARS) (pink) and DMS (blue) structural content of different RNA types (Ensembl). **d** Semantic grouping of gene ontology terms associated to the least and most structured RNAs (100 less structured (LS) vs. 100 high structured (HS) transcripts) using cleverGO. **e** Through the analysis of individual RNAs (Figs. 1 and 2b) we found that the structural content is linked to the number of partners and function of an RNA. Our analysis indicates that functionally related RNAs have similar structural content (Fig. 2c). The structure-driven protein interactivity is an intrinsic property associated with the RNA that can be traced at any regulatory level. **f** Each row shows the catRAPID interaction propensities caused by removing a physicochemical property[13,32]. The removal of α-helix (Chou) and polarity (Grantham) reduce the ability to distinguish between HS and LS ($p$ values estimated with KS test). **g** multicleverMachine analysis of the physicochemical properties of three RBP sets and proteins annotated in UniProt as binders of double-stranded RNAs (DS) or single-stranded RNAs (SS) (see Methods). 'Disorder propensity' and 'α-helix' are the properties showing significant difference and opposite results between DS and SS binders for at least two RBP databases (blue or pink indicate that DS or SS are enriched or depleted; yellow indicates no significant differences between the sets). In **b**, **c**, the boxes show the interquartile range (IQR), the central line represents the median, the notches the 95% confidence interval of the median, the whiskers add 1.5 times the IQR to the 75 percentile (box upper limit) and subtract 1.5 times the IQR from the 25 percentile (box lower limit). S.d. is shown

is polarity, which is enriched in structurally disordered proteins[49] and anti-correlates with hydrophobicity that is key in macromolecular recognition (Supplementary Table 2)[50]. As for the α-helical propensity, we note that the helices are the most frequent structural elements involved in the formation of contacts with double-stranded regions and occur in dsRBD and zinc fingers[29] (Supplementary Table 3). Our observation suggests a possible co-evolution between proteins and RNAs: while the RNA adopts complex shapes to expose binding regions, proteins change their structural content. In agreement with the key lock theory[51], we propose that natural selection favours highly structured RBPs as interactors of dsRNAs.

We validated the importance of protein polarity and helical structure by comparing three datasets of well-studied RBPs (human and yeast)[52–54] and two sets of proteins retrieved from UniProt (all organisms) as either exclusively ssRNA binders (453 proteins) or dsRNA binders (390 proteins; Supplementary Data 4). Analysis of biophysical properties with the cleverMachine approach[55] revealed that ssRNA binders and dsRNA binders differ for two properties: disorder and α-helix content (Fig. 2g). The comparison of the two sets, one against the other, indicate that RBPs interacting with highly structured RNAs are structured and hydrophobic, while disordered and polar RBPs associate with less structured RNAs (Supplementary Fig. 5). Thus, our analysis further expands what was previously reported for protein–protein interaction networks, in which structural disordered regions have been shown to play a central role[47], and suggests new rules for nucleotide base pairing with amino acids.

**RNA structure content and protein contact in chaperones.** The analysis of the human transcriptome and across organisms indicate that highly structured RNAs are prone to interact with polypeptides and, in turn, code for proteins involved in biological processes associated with large and complex contact networks. To better investigate the structure-driven protein interactivity of RNA molecules, we focused on a class of transcripts coding for proteins interacting with several partners. The natural choice for this analysis is the molecular chaperones, as they promote folding into the native state[56] and organize the assembly of phase-separate RNP assemblies[57], thus fulfilling the 'recursive' property presented in Fig. 2d. eCLIP data[30] show that most of the RNAs coding for human chaperones are involved in interactions with multiple proteins (Supplementary Fig. 6). We found a significant correlation between protein–RNA and protein–protein interactions annotated in BioGRID (Fig. 3a). This result confirms that

transcripts bound by many RBPs also code for highly contacted proteins.

To understand if the correlation between protein–protein and protein–RNA interactions is a general property or simply a feature of the chaperone family, we analysed interactions of the transcriptome ranked by PARS scores and 24 mRNAs coding for chaperones for which PARS data are available (Genecards; https://www.genecards.org; 'HSPs' set; Methods, Fig. 3b). We found a positive correlation between the amount of RNA structure and the number of BioGRID interactors of the encoded proteins (Supplementary Fig. 7a–b). Thus, our calculations agree with the GO analysis (Fig. 2d) and suggest a relationship between mRNA and their coding partners: highly structured RNAs code for highly interacting proteins.

The data presented so far suggest that RNAs related by type (e.g. miRNA, snRNA) or function (e.g. coding for chaperones) share similar structural characteristics (Fig. 2). Thus, it should be possible to estimate differences in the interaction network of two unrelated transcripts by analysing their structural content, and vice versa. To test this hypothesis, we selected the highly structured *HSP70* transcript (HS RNA, log of PARS score of −1.3 corresponding to 26% of double-stranded content, Fig. 3c) coding for a chaperone essential to regulate protein complex assemblies such as clathrin coats[58] and stress granules[22,57]. As a control we chose the RNA coding for *BRaf* that is less structured (LS RNA, score of −2.8 indicating 6% of double-stranded content according to PARS, Fig. 3c–e) and encoding for an oncogene involved in transmission of chemical signals from outside the cell to the nucleus (the structural comparison is confirmed by CROSS predictions and DMS experiments, as shown in Supplementary Fig. 8).

We found that *HSP70* has a larger number of partners (30 RBPs identified by eCLIP) than *BRaf* (9 eCLIP RBPs, 6 in common to *HSP70*, Supplementary Fig. 9), which is perfectly in agreement with the structure-driven protein interactivity property. In keeping with the trend of Fig. 1b, catRAPID indicates that proteins have a larger propensity to bind to *HSP70* than *BRaf* (Fig. 3f). Moreover, the highly structured *HSP70* codes for a protein with a higher number of interactors (244 BioGRID physical interactors), while the poorly structured *BRaf* has a protein product binding to a smaller set of molecules (88 BioGRID physical interactors). Our observations suggest that an RNA with a large number of interactions is prone to act as a network regulator: we speculate that, because of the higher interactivity, *HSP70* transcript could perform as a chaperone depending on the context.

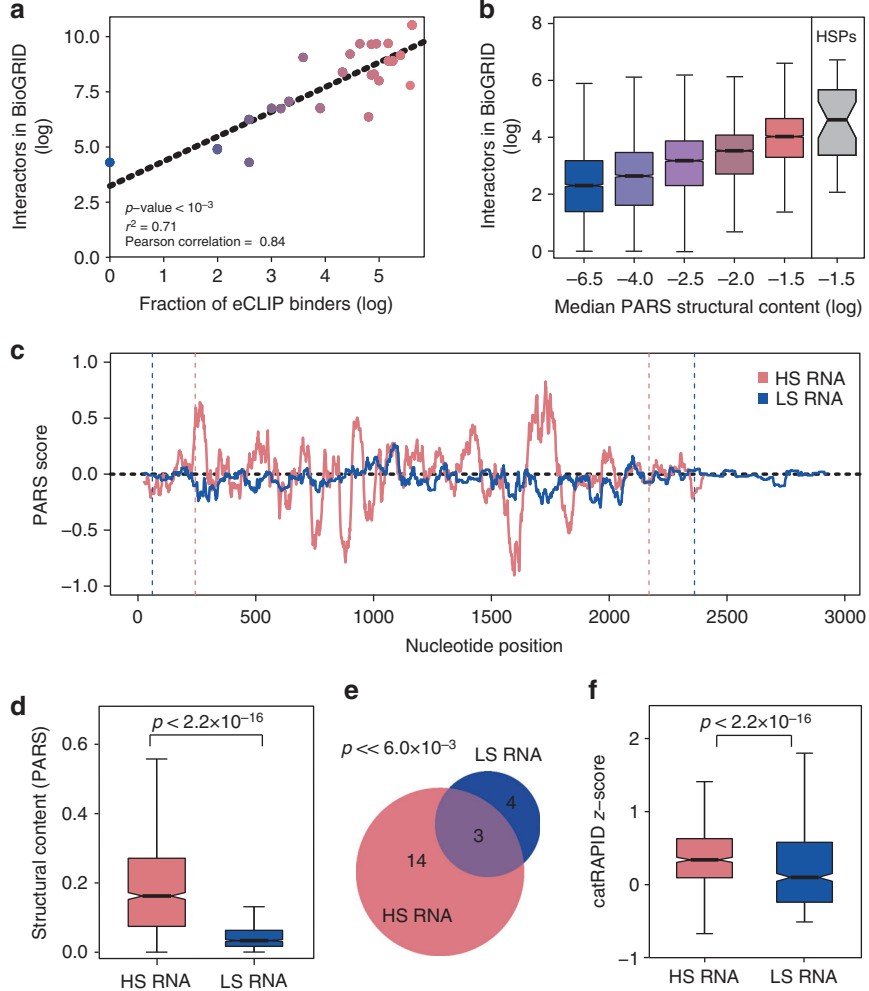

**Fig. 3** Relationship between RNA structure and protein contacts for chaperones. **a** Contacts of RNAs coding for protein chaperones, measured by enhanced CrossLinking and ImmunoPrecipitation (eCLIP)[30], and physical interactions of the corresponding coded proteins, collected from BioGRID; p value estimated with KS test. **b** Comparison between parallel analysis of RNA structure (PARS) structural content and physical interactions of the encoded proteins, collected at BioGRID, for the entire transcriptome. The transcriptome was divided in five consecutive sets containing each 20% of the transcriptome. The sets were selected regarding their PARS structural content, the range of each set from left to right are: −10.7 to −4.6; −4.6 to −3.1; −3.1 to −2.4; −2.4 to −1.9; −1.9 to −0.5. The last boxplot shows the distribution of the number of physical interactors retrieved from BioGRID for the chaperone protein family (heat-shock proteins). **c** PARS measurement of the secondary structure content of HS (HSP70, pink) and LS (BRaf, blue) transcripts. Vertical dashed lines indicate untranslated regions (UTRs). **d** PARS secondary structure content of HS and LS transcripts (p value estimated with KS test). **e** Venn diagram showing the overlap between protein interactions, measured by eCLIP, of HS and LS RNAs (empirical p value <6 × 10⁻³; estimated by comparing with the distribution of 1000 overlaps of sets sampled from eCLIP RBPs). **f** Prediction of protein binding propensity of HS and LS RNAs using catRAPID[13,32] (p value estimated with KS test). For **b**, **d**, **f**, the boxes show the interquartile range (IQR), the central line represents the median, the notches the 95% confidence interval of the median, the whiskers add 1.5 times the IQR to the 75 percentile (box upper limit) and subtract 1.5 times the IQR from the 25 percentile (box lower limit). S.d. is shown

Thus, we hypothesize that a structured RNA, because of its higher protein-interacting potential, is able to affect the protein interaction network more than a poorly structured RNA. In a proof-of-concept experiment, we used a chemical compound, biotinylated isoxazole (b-isox) to induce formation of a liquid-to-solid phase transition of a protein assembly[59,60] that we incubated with either HS (HSP70) or LS (BRaf) transcripts (Fig. 4a and Supplementary Fig. 10). We observed that HS altered the composition of the protein aggregate more than LS RNA (Fig. 4b and Supplementary Data 5). Indeed, when HS RNA was added, a significant change of concentration was observed for 29 proteins (Fig. 4c; 21 'released' set, black dots, and 8 'kept' set, red dots in Fig. 4b), while only nine proteins were identified in the LS RNA case. Thus, the composition in presence of LS RNA remained

similar to that of the background control ('static' set, grey dots in Fig. 4b).

We reasoned that the competition of RNA with the b-isox precipitate contact network[59,60] could be the result of either direct or indirect protein–RNA interactions (Fig. 5a). Yet, catRAPID predictions support the hypothesis of a direct effect: an increase in the experimental stringency (Supplementary Fig. 11; Methods) is also associated with an increase in the theoretical predictive power (Fig. 5b). In accordance with our previous analysis of RNA-binding preferences, proteins released upon HSP70 incubation result significantly deprived of polarity (Fig. 5c). Thus, our experiment suggests that the structure-driven protein interactivity of RNA molecules is active at every level, promoting individual interactions and altering the composition of condensates[12] (Fig. 2e).

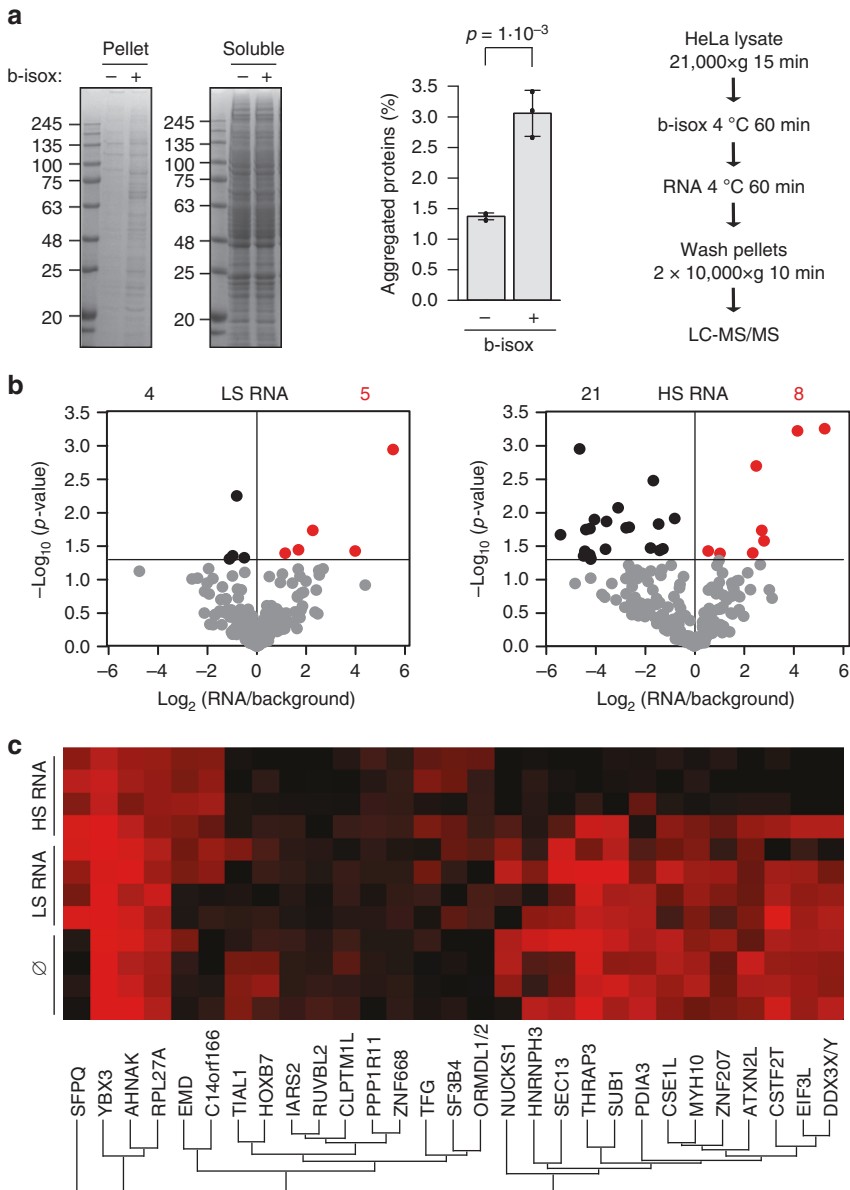

**Fig. 4** Structured RNA reduces protein aggregation in vitro. **a** Biotinylated isoxazole (b-isox)-driven aggregation of HeLa protein lysate in vitro. Left, Coomassie-stained gels, one representative experiment shown (uncropped gels are presented in the Supplementary Fig. 10). Centre, aggregated protein intensity was quantified and the difference evaluated using two-tailed $t$ test ($p = 1×10^{-3}$; $N = 3$ biological replicates shown as dots in the image). S.d. is shown. Right, experimental scheme. The aggregation efficacy was tested by comparing the resultant precipitate in the presence or absence of b-isox, this is indicated by a+ or a−, respectably. **b** Volcano plots indicate the $p$ values (Perseus measure) of the individual protein enrichments in b-isox assembly ($N = 4$ independent biological replicates). The statistical significance threshold is marked by a horizontal line (see also Supplementary Data 5). Black dots are proteins with significantly decreased concentration after the RNA incubation. Red dots are proteins with significantly increased concentration after the RNA incubation. **c** Colour-coded label-free quantitation (LFQ) intensities of proteins affected by the high structured (HS) RNA on a scale from black (low) to red (high). Hierarchical clustering by Perseus is indicated. For comparison, the LFQ intensities of the same proteins in control and in the presence of the LS RNA are plotted as well

## Discussion

Owing to recent advances in high-throughput sequencing, it is now possible to collect information on the majority of RNAs. Large-scale experiments unveiled many functions of transcripts[10,37]: chromatin modification[27], protein assembly[26] and phase separation[21], among others. At the molecular level, there are still many questions to be addressed in order to understand the full picture.

Here we focused on the relationship between RNA secondary structure and ability to interact with proteins. It is widely accepted that the structure of a molecule determines all aspects of

its life, from stability to function[10]. Yet, to the best of our knowledge, we are the first to report that the structural content of an RNA is intimately connected with the number of protein binders. We demonstrated the solidity of this observation by analysing PARS, DMS, crystals, protein microarrays and CLIP datasets, and also carrying experimental work on RNP aggregation. Our results are not completely unexpected, since lack of RNA structure is linked to more flexible and variable conformations and, thus, a shorter residence of proteins. Moreover, it should be considered that presence of a native fold favours the formation of stable and well-defined binding site that promotes

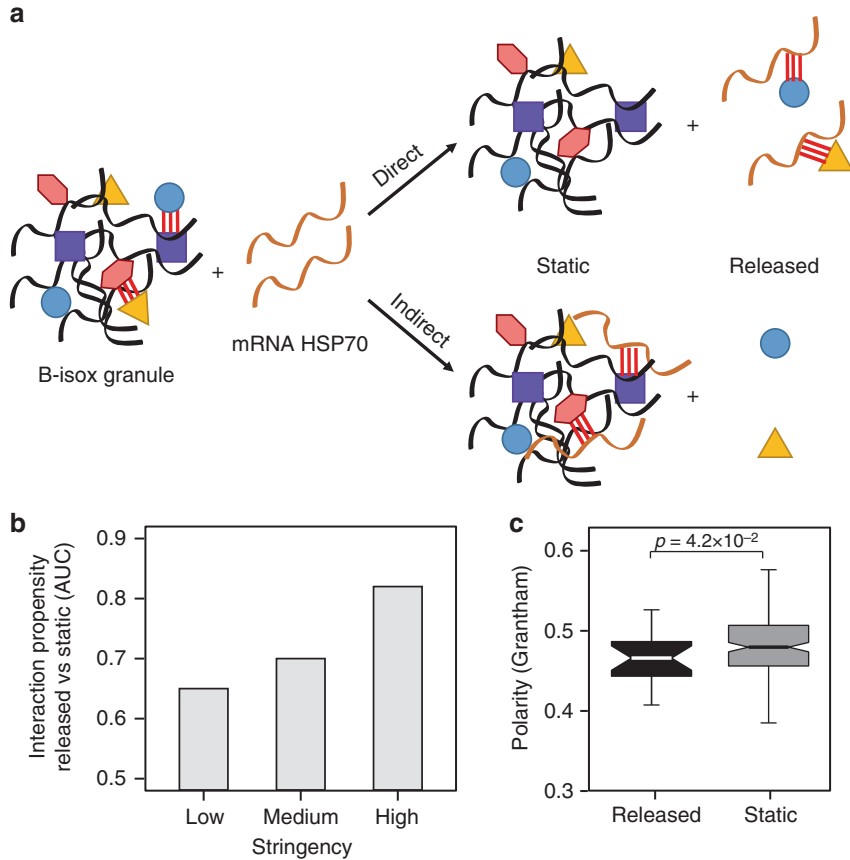

**Fig. 5** Interactions within the ribonucleoprotein condensate. **a** The release of proteins from the biotinylated isoxazole (b-isox) assembly could be the outcome of: (1) an indirect process, resulting from an interaction competition between RNA and the protein aggregate or (2) a direct process, resulting from protein sequestration by RNA. **b** catRAPID performances improve with the stringency of the b-isox experiments (Methods), suggesting a direct recruitment of proteins rescued by high structured (HS) RNA. The false discovery rate (FDR) becomes highly significant for the most-stringent experimental set (FDR = 0.1). **c** 'Released' proteins (black box) are less polar than 'static' ones (grey box), in agreement with our computational analysis ($p$ value = $4.7 \times 10^{-2}$, $p$ value estimated with KS test; see also Fig. 2f, g). Released and static proteins correspond to the black and grey dots of Fig. 4b right panel. The boxes show the interquartile range (IQR), the central line represents the median, the notches the 95% confidence interval of the median, the whiskers add 1.5 times the IQR to the 75 percentile (box upper limit) and subtract 1.5 times the IQR from the 25 percentile (box lower limit). S.d. is shown

functional roles and, in turn, evolutionary selection[12]. Thus, our findings are reminiscent of the 'target accessibility' model that links protein–RNA recognition to the secondary structure of physical contacts[61,62].

The trends presented in this work suggest the existence of a layer of regulation that directly associates an RNA with its protein product[20,48]. In agreement with our GO ontology analysis, it has been previously reported that the transcripts with high conserved secondary structure are enriched in regulatory processes in plants[63]. Indeed, a tight relationship exists between the number of protein contacts of transcripts and the participation of encoded proteins in the network, which reveals an important level of transcriptional regulation[37] for highly connected genes[47]: proteins involved in many interactions are encoded by RNAs bound by several proteins. Our observation indicates an important functional connection, since highly contacted proteins participate in many cellular processes and require tight control at the post-transcriptional level[12,64,65].

By studying transcripts sharing high sequence similarity, such as HBG1 and HBG2, we found that the structure-driven protein interactivity is finely regulated by the exact position of nucleotides. Through the study of a phase-separated RNP assembly we proved that a highly structured RNA such as *HSP70* is able to transform the interaction network by competing with pre-existent protein interactions. The main effect observed was the release of proteins from the aggregate; proteins computationally and experimentally tested to be direct interactors of *HSP70*.

While the role of HSP70 protein as a protein chaperone is well documented and there are reports on its binding to hydrophobic peptide domains to prevent aggregation and facilitate protein folding[66], very little is known about the property of the mRNA. Our data suggest an intriguing activity of the *HSP70* transcript as a chaperone solubilising the protein assembly, which creates a connection between RNA and protein activities. Our findings are in agreement with previous reports indicating that RNAs are directly involved in RBPs assembly[21]. Indeed, ribosomes have been shown to be powerful co-factors aiding the folding of polypeptide chains as they emerge from their channel[15]. Thus, we speculate that other RNAs can act as chaperones assisting the assembly of proteins. As each transcript is continuously handed off from one protein to another, we expect a mutual chaperoning effect of proteins on RNA and RNA on proteins, which is likely the result of the co-evolution between the two molecules[67].

In the future, kinetic analyses tracking the RNA–protein association will be needed to further elucidate to which extent protein partners actively contribute to RNA structure formation. Our findings are reminiscent of the lock-and-key model in the field of enzymology[51]: the structure of both, enzyme and substrate, are key determinants of their association. Yet, structure contributions are not trivial in the case of RNP associations

because the combination of different nucleotides bears an obvious specificity-determining potential. While unfolded regions promote protein–protein assembly and disordered proteins exploit short motifs to ensure high connectivity[68], the reduced nucleotide alphabet and its complementarity suggest that nature favours structure to connect RNAs with proteins[12].

The observations presented here, from transcriptome-wide to single molecule analyses, indicate that RNA controls gene regulation at multiple levels. The correlation between RNA structure and number of protein interactions could lead to the discovery of RNA functions that are presently unknown. As demonstrated in the case of protein aggregation, there are RNA-based mechanisms that control phase separation and could be important for the formation of membrane-less organelles. Overall, the complexity and diversity of protein–RNA networks reported here open the avenue for the investigation of regulatory processes.

## Methods

**RNA secondary structure measured by PARS**. To profile the secondary structure of human transcripts, we used PARS data[8,69]. PARS distinguishes double- and single-stranded regions using the catalytic activity of two enzymes, RNase V (able to cut double-stranded nucleotides) and S (able to cut single-stranded nucleotides)[2,3]. Nucleotides with a PARS score higher than 0 indicate double-stranded conformation, while values lower than 0 are considered single stranded[8,69]. Transcripts with all nucleotides undetermined were discarded from our analysis. To measure the PARS structural content, for each transcript, we normalized by length and computed the fraction of double-stranded regions over the entire sequence. Given the stepwise function $\vartheta(x) = 1$ for $x > 0$ and $\vartheta(x) = 0$ otherwise, we computed the fraction of structured domains as:

$$\text{PARS structural content} = 1/l \sum_i^l \vartheta\left(\log_{10}\frac{V(i)}{S(i)}\right), \quad (1)$$

where $V(i)$ and $S(i)$ are the number of double- and single-stranded reads. The top (larger fraction of double-stranded nucleotides) and bottom (larger fraction of single-stranded nucleotides) transcripts are listed in Supplementary Data 1.

To measure the secondary structure content of the human transcripts without the 5′- and 3′- UTR, we retrieved the corresponding locations of the 5′- and 3′-UTR from Ensembl database (section Database of RNA types) and repeated the same procedure described above eliminating the PARS values corresponding to the UTRs.

**RNA secondary structure measured by DMS**. We obtained secondary structure data from in vitro DMS modification published by Rouskin al[9]. The number of reads of each transcript was normalized to the highest value (as in Rouskin et al.[9] Fig. 2a) and then averaged. Transcripts with available DMS data for <10% of the entire sequence length were excluded from the analysis.

**catRAPID predictions of protein–RNA interactions**. The full list of protein–RNA interactions calculated with catRAPID is available in the RNAct database[31]. These interaction predictions were calculated over several months on a shared set of 80 HP BL460c nodes with two Intel Xeon E5-2680 2.70 GHz CPUs and 120 GB of usable DDR3-1600 memory each, using eight cores per cluster job, corresponding to 120 years of calculations in a high-throughput cluster. catRAPID omics was used to compute the interaction propensities of 100 LS and HS against the human RNA-binding sub-proteome[32]. catRAPID omics uses large pre-compiled protein libraries and ranks interactions based on the propensity score, with strong predictive power as well as presence of motifs and RBDs[13]. We used catRAPID omiXcore[70] to calculate the interaction propensities of *HSP70* mRNA with the ad hoc set of proteins present in the b-isox precipitate.

**RPISeq predictions of protein-RNA interactions**. RPISeq[11] is a sequence-based predictor of protein–RNA interactions based on two distinct classifiers, SVM and RF. RPISeq is available only through webserver submission and for a limited number of inputs (1 RBP against 100 transcripts per batch run). To circumvent this practical problem, we used CD-HIT (similarity clustering threshold of 85%) to select a representative group of 50 distinct RBPs out of our pool of 579 RBPs. Similarly, we built a set of 100 transcripts (ranked by PARS scores and divided into two groups, as reported in Results, 'Highly structured RNAs bind a large amount of proteins': highest values or HS, and lowest values or LS). Using both RPISeq approaches (RPISeq-SVM and RPISeq), we predicted that HS has higher interaction propensities (RPISeq-SVM: HS median: 0.80, LS median: 0.70; RPISeq-RF: HS median 0.71, LS median: 0.54; Supplementary Table 2). We estimated the interaction propensities on the same dataset using catRAPID and observed a similar trend (HS median: 0.02; LS median: −0.17; Supplementary Fig. 1a).

**ENCODE database of protein–RNA interactions**. We used human protein–RNA interactions that were identified through eCLIP experiments in two cell lines, K562 and HepG2[30], dataset downloaded 5 February 2018. The dataset contains interactions for 118 unique RBPS: 92 proteins in the K562 cell line and 76 in the HepG2 cell line. We used input-normalized eCLIP data (SM input normalization). We processed the eCLIP data using bioconductor package for R (https://www.bioconductor.org/) and mapped the genomic locations to UCSC transcript Identifiers.

To measure the fraction of protein binders for each transcript, we applied stringent cut-offs [−$\log_{10}(p$ value) >5 and −$\log_2$(fold_enrichment) >3] as in a previous work[30]. The interactions retrieved after applying the above thresholds represent the top 5% of all interactions in the eCLIP database. The number of protein binders for the transcripts at the top and bottom PARS are listed in Supplementary Table 1.

**RNA–protein interactions retrieved from POSTAR database**. iCLIP, PAR-CLIP, and HITS-CLIP interaction data were obtained from the POSTAR database (http://lulab.life.tsinghua.edu.cn/postar/) using POSTAR's thresholds ($p$ value <0.01 for Piranha, score <0.01 for CIMS, score >0.5 for PARalyzer) and mapped to human GENCODE release 27 transcripts (which directly correspond to Ensembl release 91). POSTAR provides the results of two alternative CLIP-seq data processing methods for each CLIP-seq method (two of Piranha, CIMS, and PARalyzer). We prioritized Piranha since it had been applied to all three experimental methods in POSTAR. In total, we retrieved data for 103 experiments on 85 distinct RBPs (23 studied with HITS-CLIP, 58 with PAR-CLIP and 22 with iCLIP).

**RNA secondary structure preference**. To measure the RNA structure preference of the 85 RBPs, we used DMS data (see Methods: RNA secondary structure measured by DMS) and compared the secondary structural content of the interactome of each RBP with the transcriptome as background. An RBP has (i) high structured RNA preference when the median secondary structure content of its transcriptome is higher than that of the background with $p$ value ≤0.01, (ii) 'low structured' RNA preference when the median secondary structure content of its transcriptome is lower than that of the background with $p$ value ≤0.01 and (iii) 'no preference' for the rest of the cases.

**CROSS predictions of RNA secondary structure**. We predicted the secondary structure of transcripts using CROSS (Computational Recognition of Secondary Structure[38]. CROSS was developed to perform high-throughput RNA profiling. The algorithm predicts the structural profile (single- and double-stranded state) at single-nucleotide resolution using sequence information only and without sequence length restrictions (scores >0 indicate double-stranded regions). In our analysis, we used the CROSS icSHAPE method that is trained on icSHAPE data[8].

**Protein-array experiments on protein–RNA interactions**. To find an alternative link between RNA secondary structure and the number of RBP contacts, we analysed protein-array data[36,71]. The GEO entry that we used in our analysis includes the protein interactomes of eight RNAs. For each RNA, we computed its number of interactors after applying a threshold on the signal (affinity over background >5, average of two experiments). Then, we computed the percentage structural content of the list of RNAs using CROSS. We then defined the percentage structural content as the percentage of nucleotides having a positive score (double-stranded tendency).

**PDB analysis of protein–RNA interactions**. From the PDB database we retrieved all structures that contain complexes composed of protein and RNA molecules, excluding those that contain DNA or RNA/DNA hybrid and with X-ray resolution ≤2.0 Å (exact Search Parameter: 'Chain Type: there is a Protein and a RNA chain but not any DNA or Hybrid and Resolution is between 0.0 and 2.0'). These complexes account for 196 PDB identifiers (Supplementary Table 1). For all protein–RNA pairs reported in those complexes, we consider as internal contacts the total number of times that two phosphate atoms of two distinct nucleotides are in a distance <7 Å and as external contacts the total number of times that a phosphate atom and a carbon A atom are in a distance <7 Å. We normalized those numbers by the length of the nucleotide and protein sequence:

$$\text{Internal contacts} = \log_2\left(\frac{1 + \sum \text{contacts}_{\text{internal}}}{\text{Length}_{\text{RNA}}^2}\right), \quad (2)$$

$$\text{External contacts} = \log_2\left(\frac{1 + \sum \text{contacts}_{\text{external}}}{\text{Length}_{\text{RNA}} * \text{Length}_{\text{protein}}}\right). \quad (3)$$

**Features of proteins that bind ssRNA or dsRNA**. Using physicochemical properties, the multiclever Machine[55,72] approach discriminates ssRNA and dsRNA binders from RBPs. The datasets employed in this analysis comprise 453 ssRNA binders and 390 dsRNA binders, as well as three different RBPs sets. For each feature, enrichment or depletion of a set is indicated with a specific colour:

green indicates that ssRNA or dsRNA binders are enriched with respect to the general RBP sets; red means depletion with respect to the general RBP sets; yellow indicates no significant differences between the sets ($p$ values $<10^{-5}$; Fisher's exact test).

For sets and results see Data availability section.

**HBG1 and HBG2 sequence analysis.** We used the CD-HIT algorithm (http://weizhongli-lab.org/cd-hit/) with similarity threshold of 85% to cluster RNAs for which DMS (see Methods: RNA secondary structure measured by DMS) and eCLIP data are available. CD-HIT returned 22 clusters (total 55 transcripts) meeting the criterion that a cluster should be populated with at least two transcripts. Pearson's correlation was estimated for each individual cluster. HBG1 and HBG2 were the two transcripts sharing the highest similarity (99.31%). HBG1 was found to be significantly more structured (DMS score: 0.04) than HBG2 (DMS score: 0.07) and in agreement with our observations HBG1 has more eCLIP interactors than HBG2 (29 and 14, respectively).

**Database of RNA types.** The gene type information for all RNAs that PARS value was available was extracted from the Ensembl database version GRCh38.p12 (https://www.ensembl.org/index.html).

**Human chaperones.** We retrieved the heat-shock protein members from Genecards database (www.genecards.org) searching for keyword criteria: 'HSP heat shock'. This returned as a result 1258 entries. Out of these results, we looked for the term 'Heat Shock Protein' in the description and removed entries referring to 'DNA Heat Shock Proteins', entries described as 'Pseudogenes' and 'Heat Shock Transcription Factors'. After the above criteria, the dataset was reduced to 27 gene names, and for each of those we retrieved from Ensembl, the corresponding UCSC identifier. As described (see Methods, ENCODE database of protein–RNA interactions (eCLIP)), our compiled database for eCLIP is annotated with UCSC identifiers. We used the UCSC identifiers of the HSP family to retrieve the fraction of Proteins that they bind.

**cleverGO.** We used the cleverGO algorithm[35] to analyze GO terms. The algorithm provides dynamically organized visualization of the GO terms and grouping depending on the strength of their internal connections. Separate analyses are generated for biological process, molecular function and cellular component ontologies.

Parameter of the GO semantic grouping:
Similar strength threshold: 0.5
Minimal precision: 0.6
Minimal level: 0.3
$P$ value cut-off: 0.01
For sets and results see Data availability section.

**BioGRID protein–protein interactions.** Protein–protein interaction information was retrieved from BioGRID, version 3.4.163, which contains 312,474 human non-redundant interactions (https://thebiogrid.org/). For each protein of interest we counted the number of unique interactors (physical) as defined by their Gene Name.

**Statistical analysis.** To assess whether ssRNAs exhibit different trends from dsRNAs, we used the Wilcoxon's test (Mann–Whitney $U$ test). Wilcoxon's is a non-parametric test used to compare the mean of two distributions without any given assumption about them. To compare properties and measure the difference between RNA sets, we used the KS test. KS test is also a non-parametric test used to compare the distance between two cumulative distribution functions.

**b-isox precipitation of HeLa lysate and MS.** For in vitro transcription, linearized plasmids were purified by phenol/chloroform/isoamylalcohol extraction and ethanol precipitation. Ampliscribe T7 high Yield kit (Epicenter, Madison, WI) was used to transcribe 1 µg of template for 4 h at 37 °C. The reaction mix was digested with DNaseI and RNA was purified using MegaClear transcription clean-up kit (Thermo Fisher Scientific). About 30 million HeLa cells were then trypsinized and collected. They were washed two times with ice-cold phosphate-buffered saline, resuspended in 300 µL lysis buffer (20 mM Tris, 150 mM NaCl, 0.5% NP-40, 0.1 mM PMSF, 10% glycerol, phosphatase inhibitor, RNasin (Promega) 1:100) and incubated at 4 °C for 20 min with gentle shaking. The lysate was centrifuged at 21,000 × $g$, at 4 °C for 15 min and the supernatant was collected. Protein concentration was measured using Bradford reagent and BSA standards and normalized to 2 µg/µL with lysis buffer. B-isox (10 mM stock in dimethyl sulfoxide) was added to the lysate at 100 µM and the mixture was incubated at 4 °C for 1 h with gentle shaking. After 1 h, the lysate with b-isox was splitted into three tubes (3 × 120 µL) and 1 µg HSP70 mRNA, 1 µg BRaf mRNA or the same volume of water and incubated again at 4 °C for 1 h with gentle shaking. The aggregates were then collected by centrifugation at 10,000 × $g$ at 4 °C for 10 min. The supernatant was collected and 2× sample buffer was added. The pellets were washed two times by being resuspended in 50 µL lysis buffer, vortexed, kept on ice for 10 min and

centrifuged again at 10,000 × $g$, at 4 °C for 10 min. The washed pellets were resuspended in 50 µL MS buffer (4% SDS, 100 mM HEPES, pH 7.6, 150 mM NaCl) and frozen before mass spectrometry analysis (or in 50 µL 2× sample buffer for sodium dodecyl sulfate-polyacrylamide gel electrophoresis and Coomassie blue staining). Subsequently, the samples were processed according to the FASP protocol using 30k filtration units (MRCF0R030, Millipore) for data-dependent LC-MS/MS analysis with a Q Exactive Plus mass spectrometer.The MS data were analyzed using the software environment MaxQuant version 1.5.3.30. Proteins were identified by searching MS and MS/MS data against the human complete proteome sequences from UniProtKB, version of November 2015, containing 70075 sequences. Carbamido-methylation of cysteines was set as fixed modification. N-terminal acetylation and oxidation of methionines were set as variable modifications. Up to two missed cleavages were allowed. The initial allowed mass deviation of the precursor ion was up to 4.5 ppm and for the fragment masses it was up to 20 ppm. The 'match between runs' option was enabled to match identifications across samples within a time window of 2 min of the aligned retention times. The maximum false peptide and protein discovery rate was set to 0.01. Protein matching to the reverse database or identified only with modified peptides were filtered out. Relative protein quantitation was performed using the LFQ algorithm of the Maxquant with a minimum ratio count of 1. Bioinformatic data analysis was performed using Perseus (version 1.5.2.6). The proteins with minimum three valid LFQ values in at least one group (background/Hsp70 mRNA/ B-Raf mRNA) of 4 biological replicates were considered as quantified and used for downstream analysis. Proteins significantly changed in the presence of Hsp70 or BRaf mRNA compared to the background control (water) were identified by two-sample t-test at a p cutoff of 0.05.

**b-isox contact network and catRAPID predictions.** catRAPID omiXcore integrates the interaction propensities of individual protein and RNA fragments into a unique score that was used to compare predictions with experiments. To this aim, we computed the AUC ROC of two protein groups: (i) the positive set was comprised of all proteins that were significantly released in the presence of HS mRNA (HSP70; $-\log(p$ value) >1.3 and $\log$(RNA/Background) <0) and not significantly released in the presence of LS mRNA (BRaf; $-\log(p$ value) <1.3 or $\log$(RNA/ Background) >0); (ii) as negative set we used all the proteins that did not show any significant change in the presence of either HS or LS mRNAs (HSP70 or BRaf, respectively; $-\log(p$ value) <1.3) in the b-isox precipitate.

We used a cut-off to define the experimental stringency: (i) highly stringent cases, all those proteins with minimum fold change in the presence of both the two mRNAs ($-\log(p$ value) <1.3 and $|\log$(RNA/Background)$|$ <0.15), (ii) medium stringent cases, those that show modest change ($-\log(p$ value) <1.3 and $|\log$(RNA/ Background)$|$ <0.30) and (iii) low stringent cases, those that show greater change ($-\log(p$ value) <1.3 and $|\log$(RNA/Background)$|$ <1).

We built the ROC curves and computed the corresponding AUCs for the three different cases (low stringent, medium stringent, high stringent) and we observed higher performances when increasing the experimental stringency.

**Reporting summary.** Further information on research design is available in the Nature Research Reporting Summary linked to this article.

## Data availability

The data that support the findings of this study are available from the corresponding author upon request. The MS data (Fig. 4) are available via ProteomeXchange with identifier PXD011751. In the figshare repository with https://doi.org/10.6084/m9. figshare.c.4505759.v2, we report: (i) the physicochemical properties identified by multicleverMachine[55,72] to discriminate ssRNA and dsRNA binders from RBPs; (ii) the physicochemical properties selected by the cleverMachine[55] that discriminate ssRNA and dsRNA binders; (iii) the sets and results for the GO and term analysis obtained with cleverGO[35].

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

## Acknowledgements

We thank all members of the Tartaglia's, Vabulas' laboratory, Dr. Edoardo Milanetti for the analysis of crystal structures and Dr. Danny Incarnato for insights into high-throughput approaches. The research leading to these results has been supported by European Research Council (RIBOMYLOME_309545 to G.G.T. and META-META_311522 to R.M.V.), Spanish Ministry of Economy and Competitiveness (BFU2014-55054-P and BFU2017-86970-P), 'Fundació La Marató de TV3' (PI043296) and the collaboration with Peter St. George-Hyslop financed by the Wellcome Trust. We acknowledge support of the Spanish Ministry of Economy and Competitiveness, 'Centro de Excelencia Severo Ochoa 2013-2017'. We also acknowledge the support of the CERCA Programme/Generalitat de Catalunya and of Spanish Ministry for Science and Competitiveness (MINECO) to the EMBL partnership.

## Author contributions

G.G.T. conceived the study with the help of N.S.d.G. and R.M.V., A.A. performed the calculations, M.A. and G.C. performed and analysed the MS the experiments. N.S.d.G., R.G.-M., G.G.T. and A.A. analysed the data. N.S.d.G., R.M.V. and G.G.T. wrote the manuscript.

## Additional information

**Competing interests:** The authors declare no competing interests.

