## [Peer Review File · Nature Communications]

Reviewers' comments:

Reviewer #1 (Remarks to the Author):

The manuscript by Groot NS et al connects RNA structure with an RNA's ability to interact with RNA binding proteins and suggests that structure to be a property that underlies an RNA molecule's ability to form intermolecular interaction networks. The work is interesting, although it is fairly superficial and more detailed experiments needs to be carried out to dissect the role of structure with protein interactions.

Major comments

1) The author primarily uses the PARS dataset as the experimental structure data to identify structured and unstructured transcripts in the transcriptome. There have been numerous other in vivo RNA structure data sets, generated by methods such as ic-SHAPE, SHAPE-MaP and DMS-seq. The authors should show that their conclusions hold with at least one other orthogonal structure probing method.

2) This reviewer disagree with the author's conclusions that mRNAs are the most structured RNAs in the cell, and are more structured than lncRNAs. Numerous other high throughput RNA structure papers have shown that lncRNAs tend to be more structured than mRNAs and are of intermediate structuredness between mRNAs and rRNAs – agreeing with the fact that many lncRNAs can act as molecular scaffolds. The authors should look into their analysis more to find the source of discrepancy between their analysis and that of the published literature.

3) The authors used the top 100 most structured and 100 least structured transcripts in the PARS dataset. Did the authors have a minimal cutoff for transcript abundance in these 2 classes? Also, 100 transcripts is a very small number to be statistically robust given that thousands of transcripts are probed by high throughput sequencing. Do the results still hold if the authors use the most and least structured 500 genes?

4) The authors showed that the highly structured 100 transcripts bind to more RNA binding proteins than the least structured 100 genes. Is this effect really due to structure? Have the authors checked for sequence identity? Could their observations be explained by base composition? Furthermore, what would be the number of RNA binding proteins associated with a randomly selected set of 100 transcripts?

5) The authors observed that the highly structured transcripts are enriched for GO terms involving large protein complexes. What is the background set of genes that they use? Using the entire transcriptome could generate very different results from using the expressed set of genes.

6) The authors observed that removing polarity and alpha helical properties from the catRAPID algorithm abolishes the ability of the HS and LS classes to bind to different RBPs. There should be other factors beyond polarity and alpha helical formation. Can the authors show all the factors that goes into the calculation and show the systematic effect of removing the factors one by one on RBP binding?

7) Similarly, for the cumulative distribution of protein-protein contacts for the highly structured, poorly structured and chaperone classes, what would be the expected random if the authors were to randomly select 100 transcripts above a reasonable abundance?

8) The examples of HSP70 and Braf feels anecdotal. Is the structuredness/lack of structure and RBP a

general property whereby the identity of the RNA sequence does not matter? To truly establish that it is structure that is causing changes in an RNA's ability to interact with other RNA binding proteins, the authors should mutate highly structured elements, while preserving dinucleotide content, and show that the RNA no longer interacts with as many RNA binding proteins, and vice versa.

Minor comments

1) The authors should check the manuscript for grammar and smoothness in writing.

Reviewer #2 (Remarks to the Author):

In the manuscript entitled "Insights into the structure-driven protein interactivity of RNA molecules" the authors put forth the hypothesis that highly structured RNAs are more likely to interact with protein. The authors present several lines of evidence for this hypothesis, however I have severe doubts concerning many of these lines and in particular the potential for algorithms trained on limited sets to bias the results.

Major Concerns:

1. Many of the algorithms used to predict properties here were trained on specific datasets and therefore are pre-disposed to give the observed result. My understanding is that CATRapid omics is a prediction tool that utilizes "physio-chemical properties" to predict RBP binding sites. In my reading of the publication describing this tool is primarily intended to identify protein regions that interact with RNA, not RNA regions that interact with proteins. Tracing the origin of RNA site prediction to Bellucci et al. 2011, the underlying catRAPID algorithm was trained on 592 RNA-protein pairs in the PDB at the time. It is unclear exactly what parameters go into this, however in this publication the authors note that "secondary structure propensities account for 72% of the catRAPID ability to predict protein-RNA interactions, followed by hydrogen bonding (58%) and van der Waals (26%) contributions. Occurrence of hairpin loops in nucleotide sequences and presence of helical elements in polypeptide sequences positively correlated with interaction propensities." Thus, their algorithm is predisposed to identify structured regions, as previously noted in a prior publication.

In particular, the ribosome makes up a large proportion of RNA/proteins in the pdb and is a highly structured RNA element. To demonstrate a truly robust result the authors should use other algorithms besides their own to do the predictions and not ones trained almost exclusively on RNA-protein pdb structures which will favor structured RNA as it will crystallize more easily.

2. The eCLIP data utilized for this study is documented to have artifactual positives within structured RNAs. From Nostrad et al. 2016 Nature Methods "Other regions showed even higher rates of these likely false positive CLIP signals: 86% of clusters mapping to transcripts encoded on the mitochondrial chromosome and 90% of those overlapping snoRNAs were in fact depleted in RBFOX2 eCLIP relative to SMIInput (Fig. 4a). Performing similar SMIInput-normalization for all 102 experiments, we observed that the identification of CLIP-depleted clusters within mitochondria-encoded RNA molecules and many classes of ncRNA (including snoRNA, snRNA, and rRNA) was consistent across many data sets (Fig. 4b)."

3. It is unclear where the authors get the "strength of protein-RNA contacts" from. As far as I am aware eCLIP data does not directly provide this (none of the figures in the eCLIP paper display binding constants). Also, the authors seem to frequently conflate affinity with the number of protein binding partners.

4. The entire section labeled “disorder and alpha helix distinguish between double and single stranded RNAs” is basically an artifact of the dataset that catRAPID was trained on. Most of the conclusions reached here were noted in the various catRAPID publications.

5. The authors postulate that the mRNAs which encode proteins with many contacts (e.g. chaperones) are more structured and thus interact with more RNA binding proteins. It is unclear to me why this should be the case, or what exactly the biological implications might be.

6. In the final experiment the authors demonstrate that structured transcripts are better at altering composition of stress-granules. It is not surprising to me that structures vs. unstructured transcripts will have a different impact on stress-granule concentration, as their surface chemical properties are likely to be quite distinct. So the purpose of this demonstration is unclear to me.

Reviewer #3 (Remarks to the Author):

The authors present a study of RNA-protein interaction in which the main observation is that increasing amount of RNA structure on a transcript increase or bind more proteins. The authors certainly address a highly relevant problem as RNA structure often is ignored as a player for example in transcriptome studies. Although the authors' work are greatly appreciated, there are a number concerns.

(1) A main concern is the very limited number of transcripts this analysis is based on as well as how they are selected (in particular Lines 90-92 in file 178682_0_merged_1534147925.pdf) For selected transcripts, the authors needs to:

(a) justify that these transcripts non-redundant, e.g. neither close paralogues nor represent different isoforms in the same gene

(b) justify that these transcripts representable for the entire transcriptome

(c) make sure that the more structured ones are not due to high GC content

(d) compare the average structured transcript, e.g. how do we know that the 100 least structured transcripts according to PARS are not outliers? In fact this could be one interpretation of the first peak of the bi-modal distribution in fig 1a. One can therefore with reason argue that it is the “tail” of the distribution with the right most peak the transcripts should be selected from in the range of approximately -13 and equal size area as the top end. The authors should repeat the analysis for this region as the LS set.

(e) take the FDR of PARS in account and estimate if this (and possibly how) impact the conclusions.

(f) Consider if the sharp tale of the in the low end of fig 1a, could that represent transcripts that are strongly covered by proteins and therefore show up on PARS as unstructured?

(g) Give in concordance with (b) a rationale for selecting just 100 sequences in each and in addition repeat the analysis for larger amount of selected transcripts in the LS and HS groups as in total 200 transcripts compared to e.g. GENCODEs ~59000 coding and non-coding genes and 206694 transcripts, rather small. The analysis should be repeated for group sizes of 250, 500, 1000, 2000 and 5000. The difference in structure preferences should still be seen although likely with less strength.

(2) It remain unclear if the 100 transcripts selected in each end of the spectrum are equally represented by:

(a) non-coding and coding genes

(b) which of these groups share the same length (obviously longer sequences might have a bigger chance by being match by PARS data by random

(c) whether the coding sequences respectively have the same size UTRs both 5' and 3' independently, but also the total 5'+3' UTR length per gene. If that makes up a bigger fraction of the gene it will be structured.

(d) These aspects should also be considered for the additional analysis in 1(f).

(3) For the criteria of a transcript being structured, as far as I understand, is calculated in terms of PARS matches on a given transcript. However, the RNAs may well function through domains and the number of such domain be the same although some can systematically be larger than others. The authors should provide a brief discussion of this.

(4) Whereas the probing experiments provide information of a single organism, it does not provide information about evolutionary conservation, which typically indicates function. As it is well known that a random sequence also fold in structure (e.g. more GC rich ones), the authors should take existing screens for evolutionary conserved RNA structures. In this regard, an in silico study from last year by Seemann et al, Genome Res, 2017, showed enrichment of CLIP-data over such predicted conserved RNA structures. The authors should as a minimum include a discussion of how conservation of RNA structure might further increase confidence in the results or argue that their results are not related to the evolution of RNA structure at all.

(5) The Seemann study also show that coding genes are enriched for structures up- and down-streams, and that there are fewer structures on e.g. long ncRNAs. Consistent with this and the question raised above, it remain unclear if the authors here take UTR regions into account separately and whether the "structuredness" can in particular can attributed that the 100 HS sequences have UTRs that are enriched for structures whereas the entire genes.

(6) For line 61-64 (in file 178682_0_merged_1534147925.pdf)

The authors should mention some more other examples indicated that this is wide-spread and not "only" constrained to the translation machinery.

(7) For the statement about snoRNAs in line 66 (in file 178682_0_merged_1534147925.pdf)

These authors should know that snoRNAs are not necessarily structured. H/ACA are, but far from all CD which readily can be extracted from the Rfam database. In fact this even supported by the authors *own* results where snoRNA is in the low end in Fig 2b. The authors should revise accordingly.

(8) For the usage of catRAPID the authors should

(a) take the estimated false discovery rate of catRAPID into account and elaborate on how (if any) this impact the results (line 95 in the file 178682_0_merged_1534147925.pdf).

(b) provide (line 178 in the file 178682_0_merged_1534147925.pdf) a sufficient background to the parameters (relevant here) of catRapid so readers not familiar with the program can follow you.

(c) Also add a rationale as to why these parameters are relevant to look at and least bring the accuracy of catRapid into the context of usage and possible impact on the interpretation of the results.

(9) Line 110 for "It is worth to mention that the eCLIP assays favour detection of single-stranded 110 (SS) RNA at the expense of double-stranded (DS) RNA." Please add a reference.

(10) For the construction of the three sets of transcripts (line 116 in file 178682_0_merged_1534147925.pdf), the authors should elaborate beyond the caption text for fig 1d including an exact definition of "high", "medium" and "low". If this is what is already described in Methods "RNA Secondary structure content and eCLIP", then make an explicit pointer. Possibly

substantiate.

(11) For the conclusion drawn (lines 170-171 in file 178682_0_merged_1534147925.pdf) the authors need of substantiate how such a conclusion can be drawn from using 200 transcripts.

(12) In lines 218-219 (file 178682_0_merged_1534147925.pdf) it is unclear precisely which RNAs is in which three classes and it would be helpful if the authors elaborate.

(13) The reference #10 to multicleverMachine in the Methods document (Methods.pdf) is not the correct reference of the method. Please correct.

(14) for all the methods mentioned in the main manuscript file (file 178682_0_merged_1534147925.pdf), e.g. multicleverMachine should be cited in there as well. The authors should sanitize if all methods mentioned are used.

Minor:

(15) Line 264: Perhaps "supports" instead of "demonstrates".

(16) For the programs used, e.g. catRAPID, CleverGO, multicleverMachine, etc, indicate precisely which version of the programs you used and indicate precisely all options given for how they were used in this manuscript. This should be indicated in the descriptions in the Methods section.

REVIEWER 1

- The manuscript by Groot NS et al connects RNA structure with an RNA's ability to interact with RNA binding proteins and suggests that structure to be a property that underlies an RNA molecule's ability to form intermolecular interaction networks. The work is interesting, although it is fairly superficial and more detailed experiments needs to be carried out to dissect the role of structure with protein interactions.

We thank the Reviewer for his/her interest in our work and we are grateful for the comments that allowed us to strengthen the presentation of our findings. Accordingly, new experiments have been added in the new version of the manuscript, as reported below.

- Major comments

The author primarily uses the PARS dataset as the experimental structure data to identify structured and unstructured transcripts in the transcriptome. There have been numerous other in vivo RNA structure data sets, generated by methods such as ic-SHAPE, SHAPE-MaP and DMS-seq. The authors should show that their conclusions hold with at least one other orthogonal structure probing method.

We are indeed concerned with a potential bias in the PARS technique. The reason why we originally used PARS is that it was performed on the *H. sapiens* transcriptome.

As suggested by the Reviewer, we analyzed human DMS-seq data (**Figure 1d**)¹. We found that there is a tight anti-correlation between number of protein interactions revealed by eCLIP and DMS signal (low signal indicates enrichment in double-stranded regions; **Figure R1**).

Figure R1. Transcripts establishing contacts with a large number of RNA-binding proteins (eCLIP) are more structured (i.e., less DMS-reactive). The correlation is 0.47.

For the two main cases, Hsp70 and Braf, DMS-seq provide the same results as PARS (we note that icSHAPE data are not available on these transcripts; **Supplementary Figure 8**).

Supplementary Figure 8).

To avoid potential biases caused by the use of a specific technique, we employed CROSS in our analyses (**Figure R2**; **manuscript Figure 1e** and **Supplementary Figure 8**). Moreover, our results are confirmed by analysis of crystal data (**manuscript Figure 1h**), which is independent of PARS.

Figure R2. CROSS indicates that Hsp70 (HS RNA) is more structured than Braf (LS RNA; p -value $\ll 0.01$ using Wilcoxon test).

- 2) This reviewer disagree with the author's conclusions that mRNAs are the most structured RNAs in the cell,

and are more structured than lncRNAs. Numerous other high throughput RNA structure papers have shown that lncRNAs tend to be more structured than mRNAs and are of intermediate structuredness between mRNAs and rRNAs – agreeing with the fact that many lncRNAs can act as molecular scaffolds. The authors should look into their analysis more to find the source of discrepancy between their analysis and that of the published literature.

We agree with the Reviewer that we should not make any claim on the structure of lncRNAs. Indeed, there is an ongoing debate on this class of RNAs² and a recent work from our group³ (now out at <https://www.frontiersin.org/articles/10.3389/fmolb.2018.00111/abstract>) indicates a degree of structural conservation.

As also recommended by Reviewer 3, we compared our results with those reported in Seeman *et al.* on RNA structure conservation⁴. Our conclusion is that lncRNAs are a heterogeneous class of transcripts. Some lncRNAs (*Xist*, *Neat* as well as poly-nucleotide expansions⁵) are structured and act as scaffolds in complex contact network through their domains. However, this may not be true for all lncRNAs.

To be consistent with the current bibliography we have informed the reader about possible discrepancies. With this aim we have adapted the main manuscript:

“We then wondered whether specific RNA structures are involved in protein regulation. To this aim, we divided the human transcriptome in different classes and analysed their secondary structure as detected by two independent experimental techniques, PARS and DMS. Both techniques show that protein-coding RNAs have the largest structural content (Fig. 2c, Supplementary Table 4)³⁸. Although part of the mRNA structure is concentrated in the UTRs⁸, when these are excluded, the distribution of the structural content does not change substantially (Supplementary Fig. 3; Pearson’s correlation between transcripts with and without UTRs = 0.94). The RNAs known to interact with proteins such as snRNAs³⁹ and snoRNAs²⁸ show high amount of structure, whereas RNAs targeting complementary regions in nucleic acids such as antisense, miRNAs and a number of long intergenic non-coding RNAs (lincRNAs)^{40,41} feature the smallest amounts of structure⁴² (Supplementary Table 4).

*In agreement with our findings, Seemann *et al.* previously observed a tight relationship between protein binding and conservation of structural elements in mRNAs, which occur to a lesser extent in long non-coding RNAs (lncRNAs)¹³. Although lincRNAs show a lower amount of double stranded regions (lowest in PARS, third-lowest in DMS), it is important to mention that some of them, such as NEAT1⁴³ and XIST²⁷, use structured domains to scaffold protein assembly. Indeed, there is currently an active debate^{44,45} on the structural differences between coding and non-coding transcripts and our analysis of DMS and PARS data does reveal opposite results for specific RNA types, which suggests that further investigations are needed (Fig. 2c; Supplementary Table 4).”*

We also want to note that while detection of non-coding RNAs might be limited by experimental constrains, the main conclusions of our paper do not depend on their classification. We have addressed this point in the revised version of the manuscript.

- 3) The authors used the top 100 most structured and 100 least structured transcripts in the PARS dataset. Did the authors have a minimal cutoff for transcript abundance in these 2 classes? Also, 100 transcripts is a very small number to be statistically robust given that thousands of transcripts are probed by high throughput sequencing. Do the results still hold if the authors use the most and least structured 500 genes?

We selected 100+100 transcripts to allow *cat*RAPID calculations (computationally expensive: 200 transcripts

imply prediction of around 100,000 RNA-protein interactions). Following up on Reviewer's suggestion, we analysed all the available experimental data and found a significant correlation of 0.52 between RNA structure and binding contacts transcriptome-wide (**Figure R3**) (see also **Figure R1**).

With the whole transcriptome analysis (**Figure 1d-e**) we can now confirm that there is a tight relationship between RNA structure and protein binding, which *follows the trend previously reported for the 100+100 transcripts and crystallographic data (Figure 1h)*.

Figure R3. A strong correlation exists between structural content (PARS) and number of binding proteins (eCLIP), thus conforming the relationship found in crystallographic data (**Figure 1h**) and

predicted by our computational approaches (catRAPID). See also **Figure 1f**.

Importantly, RNA abundance does not correlate with number of protein interactions (**Figure R4**).

Figure R4. Transcript abundance does not predict protein-RNA interaction detection by eCLIP. **(A)** Receiver operating characteristic (ROC) area under the curve (AUC) values for individual RNA-binding proteins, generated using stringently filtered ENCODE Project eCLIP protein-RNA interactions from the K562 cell line as positives, and using transcript expression from the ENCODE Project as the continuous variable. **(B)** ROC AUC values for transcript expression from the ENCODE Project against eCLIP positives in HepG2 cells.

- 4) The authors showed that the highly structured 100 transcripts bind to more RNA binding proteins than the least structured 100 genes. Is this effect really due to structure? Have the authors checked for sequence identity? Could their observations be explained by base composition? Furthermore, what would be the number of RNA binding proteins associated with a randomly selected set of 100 transcripts?

As shown in our reply above (**Figure R3**), when the analysis is applied to the whole transcriptome, the trend is still present. Thus, the result obtained by selecting the two extreme sets was just an indication of what we can observe in a more quantitative trend (see also **Figure R5**).

Figure R5. We observe a weak relationship (correlation of 0.09) between GC and PARS structural contents, which is in agreement with previous reports indicating a contribution of GC pairing to RNA stability⁶.

Yet, we felt that there was need for an example about the role of the RNA sequence and composition at the acquisition on secondary structure and, thus, on the interactivity.

To this aim, we compared the secondary structure and protein interactions of similar transcripts. To cluster the RNAs by sequential similarity we used the CD-HIT algorithm at <http://cd-hit.org/>. With a threshold of 85% similarity, we found 22 clusters (total of 55 transcripts) with at least one protein contact revealed by eCLIP. At the most stringent threshold (sequence identify of 99%), we found two transcripts with significantly different secondary structure (DMS: HBG1=0.04 and HBG2=0.07; p-value=0.003; KS test). eCLIP experiments show that the more structured variant (HBG1) interacts with twice as much proteins (interactions: HBG1=29 and HBG2=14).

This extreme case shows the relevance of trend identified in our analysis of high-throughput data (**Figure 2b**)

We also addressed this question at the text:

*“The two transcripts sharing the highest similarity (99.31 %) are the gamma globins HBG1 and HBG2 that are expressed in fetal liver, spleen and bone marrow (NCBI Gene ID: 3048). The gamma globin variant with higher structure (HBG1) has a significantly larger number of protein interactors (HBG1, average DMS signal of 0.04, 29 interactors; HBG2, average DMS signal of 0.07, 14 interactors; p-value=0.003; KS test; **Fig. 2b**). While the nucleotide composition of the two transcripts remains nearly the same (HBG1: 148a, 136t, 148c, 152g; HBG2: 148a, 136t, 146c, 153g), the differences between HBG1 and HBG2 are concentrated in regions where the secondary structure is altered (**Supplementary Fig. 2**). These results indicate that protein interactivity is tightly associated with conformational changes in elements of secondary structure. Interestingly, the increased double-stranded content in HBG1, especially in the 3' UTR, is mirrored by an accumulation of translation regulatory elements (**Fig. 2b**) and a concomitant decrease in expression (NCBI Gene ID: 3048).”*

- 5) The authors observed that the highly structured transcripts are enriched for GO terms involving large protein complexes. What is the background set of genes that they use? Using the entire transcriptome could generate very different results from using the expressed set of genes.

The background was set to the *Homo sapiens* transcriptome (Gencode Basic). However, we repeated the GO analysis with the 25% most expressed transcripts in the eCLIP cell lines and found similar results (**Figure R6**).

Figure R6. Semantic clustering performed by cleverGO using transcripts with high expression (K562 cell line).

- 6) The authors observed that removing polarity and alpha helical properties from the catRAPID algorithm abolishes the ability of the HS and LS classes to bind to different RBPs. There should be other factors beyond polarity and alpha helical formation. Can the authors show all the factors that goes into the calculation and show the systematic effect of removing the factors one by one on RBP binding?

Accordingly, we report all the details in the main **Figure 2e**.

- 7) Similarly, for the cumulative distribution of protein-protein contacts for the highly structured, poorly

structured and chaperone classes, what would be the expected random if the authors were to randomly select 100 transcripts above a reasonable abundance?

Following up on the Reviewer's comment, we used the whole transcriptome as a background (Figure R7; black dashed line). Consistently with our reply to point n.3, we found that that the least structured RNAs (blue line) show lower amount of interactions, while chaperones and the highest structured RNAs (grey and red) have a larger number of interactors. We further investigated this trend and found a tight correlation between RNA structure and number of interactions of the encoded protein (Figures 3b and 3c).

Figure R7. Using the whole transcriptome as a background (black dotted line), we confirm that chaperones transcripts (grey) have higher structural content, comparable to RNAs classified as highly-structured (red).

- 8) The examples of HSP70 and Braf feels anecdotal. Is the structuredness/lack of structure and RBP a general property whereby the identity of the RNA sequence does not matter? To truly establish that it is structure that is causing changes in an RNA's ability to interact with other RNA binding proteins, the authors should mutate highly structured elements, while preserving dinucleotide content, and show that the RNA no longer interacts with as many RNA binding proteins, and vice versa.

All the analyses carried out (computational predictions, microarrays, crystallography, PARS/DMS-seq, CLIP-seq) point to the interaction-structure relationship. In this line, the beauty of our experimental work lies in its simplicity: using two unrelated transcripts having similar length we confirm that the one harbouring a higher number of double-stranded regions interacts more with proteins, thus altering the architecture of ribonucleoprotein condensates. **The finding that Hsp70 and Braf transcripts have different interaction properties has not been reported in literature before and will be used for future investigations.**

Additionally, we took up the challenge of compare sequences with minimal dinucleotide differences. To this aim, we used published data and searched for transcripts with strong sequence identity (i.e. > 99%). The properties of the top candidates (case with the highest sequence identity in the human transcriptome) discussed in **Point 4** fit perfectly with the trend observed and the sequential differences overlap with regions of secondary structure.

We think that by adding this new analysis to the text we have further increased the strength of our arguments.

REVIEWER 2

- In the manuscript entitled “Insights into the structure-driven protein interactivity of RNA molecules” the authors put forth the hypothesis that highly structured RNAs are more likely to interact with protein. The authors present several lines of evidence for this hypothesis, however I have severe doubts concerning many of these lines and in particular the potential for algorithms trained on limited sets to bias the results.

We thank the Reviewer for appreciating the lines of evidence supporting our hypothesis and we are grateful for his/her comments that have been used to improve the presentation of our findings. All the points raised by the Reviewer have been addressed, as indicated below – in particular, we focused on showing that our results are not an artefact caused by computational or experimental biases.

- Major Concerns:

1. Many of the algorithms used to predict properties here were trained on specific datasets and therefore are pre-disposed to give the observed result. My understanding is that CATRapid omics is a prediction tool that utilizes “physio-chemical properties” to predict RBP binding sites. In my reading of the publication describing this tool is primarily intended to identify protein regions that interact with RNA, not RNA regions that interact with proteins. Tracing the origin of RNA site prediction to Bellucci et al. 2011, the underlying catRAPID algorithm was trained on 592 RNA-protein pairs in the PDB at the time. It is unclear exactly what parameters go into this, however in this publication the authors note that “secondary structure propensities account for 72% of the catRAPID ability to predict protein-RNA interactions, followed by hydrogen bonding (58%) and van der Waals (26%) contributions. Occurrence of hairpin loops in nucleotide sequences and presence of helical elements in polypeptide sequences positively correlated with interaction propensities.” Thus, their algorithm is predisposed to identify structured regions, as previously noted in a prior publication. In particular, the ribosome makes up a large proportion of RNA/proteins in the pdb and is a highly structured RNA element. To demonstrate a truly robust result the authors should use other algorithms besides their own to do the predictions and not ones trained almost exclusively on RNA-protein pdb structures which will favor structured RNA as it will crystallize more easily.

catRAPID was indeed trained on crystallographic and NMR data (collected in 2010) and predicts binding regions in both protein and RNA molecules. Yet, the findings identified by catRAPID are also confirmed by non-crystallographic experiments (not used for the training of catRAPID): eCLIP, and microarrays, as well as PARS and CROSS data: **Figures 1d, 1e, 1f** and **1g**, CROSS: see also **Supplementary Figure 8**). The statistical analysis of Uniprot data (manuscript **Figure 2f**) on both canonical and non-canonical RNA-binding proteins points to the same findings.

Additionally, we found a strong correlation between structural content (PARS/DMS-seq; **Figures R1** and **R3**)

and number of binding proteins (eCLIP, iCLIP, PAR-CLIP and HITS-CLIP), which supports our original claim (**Figure 1f**). Anyway, we agree with the Reviewer that an additional validation with an independent method could be useful and we performed calculations with *RPIseq*⁷ (trained on immunoprecipitation and microarrays, i.e. not on crystals) obtaining the same trends (**Figure R8**).

Figure R8. Comparing the transcripts with the highest (HS) and lowest (LS) amount of structure (100+100 RNAs), the RPIseq method (both the two RF and SVM versions) reproduces the same trend identified by catRAPID, The p-value is $< 10^{-9}$ (Kolmogorov-Smirnov test).

Our results indicate that a stable RNA is prone to establish interactions with proteins, which is not the consequence of a bias in our method but a biophysical principle.

- 2. The eCLIP data utilized for this study is documented to have artifactual positives within structured RNAs. From Nostrad et al. 2016 Nature Methods “Other regions showed even higher rates of these likely false positive CLIP signals: 86% of clusters mapping to transcripts encoded on the mitochondrial chromosome and 90% of those overlapping snoRNAs were in fact depleted in RBFOX2 eCLIP relative to SMInput (Fig. 4a). Performing similar SMInput-normalization for all 102 experiments, we observed that the identification of CLIP-depleted clusters within mitochondria-encoded RNA molecules and many classes of ncRNA (including snoRNA, snRNA, and rRNA) was consistent across many data sets (Fig. 4b).”

The statement reported by the Reviewer does not contradict our results. Anyway, our findings are not only supported by eCLIP data (now also iCLIP, PAR-CLIP and HITS-CLIP, **Figure 1f**) but protein arrays (**manuscript Figure 1g**), crystallographic data (**manuscript Figure 1h**) and other statistics (**manuscript Figure 2f**).

Notably, CLIP approaches have a strong propensity to detect single stranded RNA, as also indicated by Jernej Ule (jernejule@gmail.com) who developed iCLIP: “crosslinking requires contact to the base, so according to biophysics, should only happen if single-stranded. To be honest, I can’t find any good citation for it, but certainly the cDNA starts have strong bias for single-strandedness across all data.”

Indeed, a reference for this observation can be found in previous literature ⁸.

Thus, our findings are even more striking, given the tendency of CLIP approaches to under-represent transcripts enriched in double-stranded regions.

- 3. It is unclear where the authors get the “strength of protein-RNA contacts” from. As far as I am aware eCLIP data does not directly provide this (none of the figures in the eCLIP paper display binding constants). Also, the authors seem to frequently affinity with the number of protein binding partners.

To avoid confusion, we do not use “strength” and “affinity” in the main text. To measure the fraction of protein binders for each transcript we applied stringent cut-offs [$-\log_{10}(\text{pvalue}) > 5$ and $-\log_2(\text{fold_enrichment}) > 3$] as in a previous work ⁹.

- 4. The entire section labeled “disorder and alpha helix distinguish between double and single stranded RNAs” is basically an artifact of the dataset that catRAPID was trained on. Most of the conclusions reached here were noted in the various catRAPID publications.

Figure 2f and **Supplementary Table 5** show that the trend is present in Uniprot data (i.e., independently of catRAPID analysis or crystals in PDB database). See also our answer to Point n. 1.

- 5. The authors postulate that the mRNAs which encode proteins with many contacts (e.g. chaperones) are more structured and thus interact with more RNA binding proteins. It is unclear to me why this should be the case, or what exactly the biological implications might be.

We thank the Reviewer for raising a point about the biological relevance of our observations. However, we do not state what is mentioned above. A correct statement would be:

“We postulate that the mRNAs with higher structural content, and thus interacting with more RNA-binding proteins, code for proteins with a larger number of contacts (e.g. chaperones).”

Indeed, we (1) observe a relationship between RNA structure and protein contacts and (2) show that the RNA structure is linked to properties of the encoded protein. In other words, we connect the following facts:

- (1) The amount of RNA structure correlates with number of protein interactions (our original finding);
- (2) The RNA is regulated by RBPs in every moment of its life¹⁰;

As a direct consequence of the two points above, it can be assumed that an RNA binding to many proteins is also highly regulated^{11,12}. We note that:

- (3) Proteins involved in many interactions are able to influence several cell processes^{13,14}.
- (4) The expression of highly regulated genes is tightly controlled at the mRNA level^{15,16}.

In other words, an RNA binding to many proteins can be regarded as a highly regulated transcript and, therefore, it is expected to be involved in several interactions. Our findings are supported by a tight relationship that we identified while replying to the Reviewer: RNA structure is correlated with the number of interactions of the encoded protein (**Figure R9**), which confirms the results presented in **manuscript Figure 3c**. We have addressed this in the Discussion section:

“ Indeed, a tight relationship exists between the number of protein contacts of transcripts and the participation of encoded proteins in the network, which reveals an important level of transcriptional regulation³⁷ for highly connected genes⁴⁶: proteins involved in many interactions are encoded by mRNAs bound by several proteins. Our observation reveals an important functional connection, since highly contacted proteins participate in many cellular processes and require tight control at various post-transcriptional levels^{13,63,64} ”

Figure R9. A correlation of 0.50 exists between the structural content of transcripts and the number of physical interactions of the encoded protein reported in the BioGRID.

- 6. In the final experiment the authors demonstrate that structured transcripts are better at altering composition of stress-granules. It is not surprising to me that structures vs. unstructured transcripts will have a different impact on stress-granule concentration, as their surface chemical properties are likely to be quite distinct. So the purpose of this demonstration is unclear to me.

Indeed, the different amounts of single- and double-stranded regions change the chemical properties of the RNA surface, influencing the affinity of proteins binding. However, the consequences on ribonucleoprotein condensation are far from being trivial.

Since b-isox precipitation involves several proteins and is accompanied by appearance of amyloid-like deposits¹⁷, its use allow us to precisely evaluate the effects of RNA competition on heterogeneous interactomes. We note that interactions established during aggregation are not selected evolutionary, creating the opportunity for us to get rid of biological or functional effects of the RNA molecules added.

In a proof-of-concept experiment, we showed that two transcripts with similar length, Hsp70 and Braf have different effects on aggregation. Hsp70 is more structured than Braf (as confirmed by PARS, and DMS-seq; **Figure 3d** and **Supplementary Figure 8**) and more prone to interact with proteins in the b-isox precipitate. In essence, *our results show for the first time that the structure-driven protein interactivity of RNA molecules has an effect on the formation of biological condensates.*

REVIEWER 3

- *The authors present a study of RNA-protein interaction in which the main observation is that increasing amount of RNA structure on a transcript increase or bind more proteins. The authors certainly address a highly relevant problem as RNA structure often is ignored as a player for example in transcriptome studies. Although the authors' work are greatly appreciated, there are a number concerns.*

We thank the Reviewer for his/her great interest in our work that aims to uncover the role of RNA in protein networks. We are grateful for the comments that allowed us to strengthen the presentation of our findings.

- *(1) A main concern is the very limited number of transcripts this analysis is based on as well as how they are selected (in particular Lines 90-92 in file 178682_0_merged_1534147925.pdf) For selected transcripts, the authors needs to:*
 - (a) justify that these transcripts non-redundant, e.g. neither close paralogues nor represent different isoforms in the same gene*
 - (b) justify that these transcripts representable for the entire transcriptome*
 - (c) make sure that the more structured ones are not due to high GC content*
 - (d) compare the average structured transcript, e.g. how do we know that the 100 least structured transcripts according to PARS are not outliers? In fact this could be one interpretation of the first peak of the bi-modal distribution in fig 1a. One can therefore with reason argue that it is the "tail" of the distribution with the right most peak the transcripts should be selected from in the range of approximately -13 and equal size area as the top end. The authors should repeat the analysis for this region as the LS set.*
 - (e) take the FDR of PARS in account and estimate if this (and possibly how) impact the conclusions.*
 - (f) Consider if the sharp tale of the in the low end of fig 1a, could that represent transcripts that are strongly covered be proteins and therefore show up on PARS as unstructured?*
 - (g) Give in concordance with (b) a rationale for selecting just 100 sequences in each and in addition repeat the analysis for larger amount of selected transcripts in the LS and HS groups as in total 200 transcripts compared to e.g. GENCODEs ~59000 coding and non-coding genes and 206694 transcripts, rather small. The analysis should be repeated for group sizes of 250, 500, 1000, 2000 and 5000. The difference in structure preferences should still be seen although likely with less strength.*

The 100+100 RNAs were originally selected due to the computational cost of performing the *cat*RAPID analysis (10^6 interactions with only canonical RBPs and $>10^7$ including all proteins). To demonstrate that the observed trend is not only restricted to the selected RNAs, we decided to investigate the relationship between PARS and eCLIP at the whole transcriptome level (i.e., not only the extreme 100 cases of the RNA structure distribution) and analyze other structural information (DMS) and CLIP-seq data. All the points (a), (b), (c), (d), (f), (g) are fully addressed by the new analysis (see **Figure 1d** and **Figure 1e**).

The results are reported in **Figure R1** and **Figure R3** of this point-to-point response to the Reviewers.

As for the error associated with PARS data (e), we note that the additional DMS analysis confirms our previous conclusions (**Figure 1e**), thus indicating that the results are genuinely independent of the particular technique. For sake of completeness, we would like to mention that we used PARS data associated with $FDR < 0.1$. We also note that the GC content is expected to be higher in stable sequences, as the interaction between G and C is the most energetically favourable (**Figure R5**)^{6,18}.

These remarks are related to the comments raised by Reviewer 1 in Points 3, 4 and 7 and we refer to them for completeness.

- (2) It remain unclear if the 100 transcripts selected in each end of the spectrum are equally represented by:
 - (a) non-coding and coding genes
 - (b) which of these groups share the same length (obviously longer sequences might have a bigger chance by being match by PARS data by random)
 - (c) whether the coding sequences respectively have the same size UTRs both 5' and 3' independently, but also the total 5'+3' UTR length per gene. If that makes up a bigger fraction of the gene it will be structured.
 - (d) These aspects should also be considered for the additional analysis in 1(f).

These questions were addressed by removing the discrete analyses (see reply above and **Figure R1** as well as **Figure R3**). Regarding the UTRs, we agree with the Reviewer that the manuscripts should include a discussion about the UTRs. We found that removing the UTRs, the structural content does not change, as shown in **Figure R10 (Supplementary Figure 3)**.

Figure R10. RNA structural content with and without UTRs. We observe a correlation of 0.94 for all the transcriptome.

To address this, we have added the text below:

*“Although part of the mRNA structure is concentrated in the UTRs⁸, when these are excluded, the distribution of the structural content does not change substantially (**Supplementary Fig. 3**; Pearson’s correlation between transcripts with and without UTRs = 0.94). The RNAs known to interact with proteins such as snRNAs³⁹ and snoRNAs²⁸ show high amount of structure, whereas RNAs targeting complementary regions in nucleic acids such as antisense, miRNAs and a number of long intergenic non-*

*coding RNAs (lincRNAs)^{40,41} feature the smallest amounts of structure⁴² (**Supplementary Table 4**). In agreement with our findings, Seemann et al. previously observed a tight relationship between protein binding and conservation of structural elements in mRNAs, which occur to a lesser extent in long non-coding RNAs (lncRNAs)¹³.”*

In agreement with this finding, we note that the experiments with Hsp70 and Braf (**Figure 4**) RNAs were performed in absence of the UTRs. Removing the UTRs does not change the overall structural content (**Figure 3d** and **Supplementary Figure 8**).

Moreover, in the specific case of HBG1 and HBG2, which was added during revision (**Figure 2b**), we report:

*“Interestingly, the increased double-stranded content in HBG1, especially at the 3' UTR, is associated with in an augment in interaction of translation regulatory elements (**Fig. 2b**) and a concomitant decrease in expression levels (NCBI Gene ID: 3048).”*

- (3) For the criteria of a transcript being structured, as far as I understand, is calculated in terms of PARS matches on a given transcript. However, the RNAs may well function through domains and the number of such domain be the same although some can systematically be larger than others. The authors should provide a brief discussion of this.

We note that the structural content is normalized by the length of the transcript to avoid possible biases.

We agree with the Reviewer that the manuscript requires a discussion about the functional and structural domains and how changes in them affect the transcripts interactivity independently of the length. With this aim, we specifically analysed transcripts of similar length (range of 50-4000 nucleotides; see point below).

Transcriptome-wide, we only found a very weak correlation of 0.10 between secondary structure and RNA length (Figure R11), as now reported in the main text:

“As for the PARS data, we found a weak correlation (< 0.10 ; Pearson’s) with RNA length and GC content, indicating that these two factors positively contribute to the secondary structure by increasing the size of the conformational space as well as the overall stability³⁵.”

Figure R11. Weak correlation between structural content and transcript length.

Moreover, the properties of highly similar RNAs (Figure 2b) perfectly fit with the trend previously reported: structural content and number of protein contacts are positively correlated.

We think that by adding this new *in silico* analysis we are able to show the importance of our findings.

This question is partially related to **Point 8** of **Reviewer 1**.

- *Whereas the probing experiments provide information of a single organism, it does not provide information about evolutionary conservation, which typically indicates function. As it is well known that a random sequence also fold in structure (e.g. more GC rich ones), the authors should take existing screens for evolutionary conserved RNA structures. In this regard, an *in silico* study from last year by Seemann et al, Genome Res, 2017, showed enrichment of CLIP-data over such predicted conserved RNA structures. The authors should as a minimum include a discussion of how conservation of RNA structure might further increase confidence in the results or argue that their results are not related to the evolution of RNA structure at all.*

We fully agree with the Reviewer. The findings reported in Seemann et al fully support our observations. Indeed, the correlation between eCLIP interactions and structural conservation highlight the functional role of RNA structure. We now cite Seemann et al in different parts of the text (the number ‘13’ refers to its citation):

“Computational methods are being developed to find interaction patterns and understand physico-chemical features of the transcripts¹², their conservation between species¹³ and binding partners¹⁴.”

“In this large spectrum of activities, RNA structure dictates the precise binding of proteins by creating spatial patterns and alternative conformations and binding sites¹³.”

“The RNAs known to interact with proteins such as snRNAs³⁹ and snoRNAs²⁸ show high amount of structure, whereas RNAs targeting complementary regions in nucleic acids such as antisense, miRNAs and a number of long intergenic non-coding RNAs (lincRNAs)^{40,41} feature the smallest amounts of structure⁴² (Supplementary Table 4).

In agreement with our findings, Seemann et al. previously observed a tight relationship between protein binding and conservation of structural elements in mRNAs, which occur to a lesser extent in long non-coding RNAs (lncRNAs)¹³. “

“Moreover, it should be considered that presence of a native fold favours the formation of stable and well defined binding site that promote functional roles and, in turn, evolutionary selection¹³.”

“Our observation reveals an important functional connection, since highly contacted proteins participate in many cellular processes and require tight control at various post-transcriptional levels^{13,63,64}.”

“While unfolded regions promote protein-protein assembly and disordered proteins exploit short motifs to ensure high connectivity⁶⁷, the reduced nucleotide alphabet and its complementarity suggest that nature favours structure to connect RNAs with proteins¹³”

- (5) The Seemann study also show that coding genes are enriched for structures up- and down-streams, and that there are fewer structures on e.g. long ncRNAs. Consistent with this and the question raised above, it remain unclear if the authors here take UTR regions into account separately and whether the “structuredness” can in particular can attributed that the 100 HS sequences have UTRs that are enriched for structures whereas the entire genes.

We consider that this question is closely related to the previous (Point 2) and following (Point 6) remarks. In brief, we do report that human mRNAs are more structured in the UTR regions and expanded our analysis to the full transcriptome (see **Figure R10, R11** and **R4**). We now also cite Seemann et al to support our observations.

“ The RNAs known to interact with proteins such as snRNAs³⁹ and snoRNAs²⁸ show high amount of structure, whereas RNAs targeting complementary regions in nucleic acids such as antisense, miRNAs and a number of long intergenic non-coding RNAs (lincRNAs)^{40,41} feature the smallest amounts of structure⁴² (**Supplementary Table 4**). In agreement with our findings, Seemann et al. previously observed a tight relationship between protein binding and conservation of structural elements in mRNAs, which occur to a lesser extent in long non-coding RNAs (lncRNAs)¹³..”

Yet, we did not find a significant change in the transcriptome structure after removing the UTRs (**Figure R10**), which indicates that UTRs, although important, should not be the focus of this investigation (**Figure R12**).

Figure R12. We do not find correlation between A) 3' B) 5' UTR length and overall structural content (see also **Figures R5, R10** and **R11**).

- For line 61-64 (in file 178682_0_merged_1534147925.pdf) The authors should mention some more other examples indicated that this is wide-spread and not “only” constrained to the translation machinery.

We have added two new examples in the text:

“There are several cases of nucleotide chains acting as scaffolds for protein complexes²¹: NEAT1 structured domains attract paraspeckle components²⁶ and XIST repeat regions sequester proteins to orchestrate X-chromosome inactivation²⁷. By contrast, poorly structured snoRNAs have been shown to facilitate the assembly of other transcripts²⁸”

- (7) For the statement about snoRNAs in line 66 (in file 178682_0_merged_1534147925.pdf) These authors should know that snoRNAs are not necessarily structured. H/ACA are, but far from all CD which readily can be extracted from the Rfam database. In fact this even supported by the authors *own* results where snoRNA is in the low end in Fig 2b. The authors should revise accordingly.e

We thank the Reviewer for spotting this omission and we accordingly adapted the text to consider also the existence of low structured snoRNAs:

“By contrast, poorly structured snoRNAs have been shown to facilitate the assembly of other transcripts²⁸.”

Regarding **Figure 2**, some snoRNAs are poorly structured (**Figure 2c**), but the median is larger than that of other non-coding RNAs.

- (8) For the usage of catRAPID the authors should
 - (a) take the estimated false discovery rate of catRAPID into account and elaborate on how (if any) this impact the results (line 95 in the file 178682_0_merged_1534147925.pdf).
 - (b) provide (line 178 in the file 178682_0_merged_1534147925.pdf) a sufficient background to the parameters (relevant here) of catRapid so readers not familiar with the program can follow you.
 - (c) Also add a rationale as to why these parameters are relevant to look at and least bring the accuracy of catRapid into the context of usage and possible impact on the interpretation of the results.
- (a) The FDR of 0.05 is used in *catRAPID omics* calculations¹⁷. We note that in all our analyses we intentionally avoided the introduction of a threshold to allow comparisons between same-size groups (**Figures 1b, 3f and 5b**).
- (b) The relevant parameters are now explained in **Figure 2e** and details of the method are mentioned in the main text and linked to relevant literature²⁰.
- (c) In the original manuscript, we estimated that *catRAPID* is able to separate 78% of interacting from non-interacting protein-RNA pairs²¹ and we recently confirmed on $> 10^5$ interactions that the area under the ROC curve is 0.78²².

These points are now mentioned in the main text and materials and methods.

- (9) Line 110 for “It is worth to mention that the eCLIP assays favour detection of single-stranded (SS) RNA at the expense of double-stranded (DS) RNA.” Please add a reference.

The reference PMID: 8718690 has been added.

- (10) For the construction of the three sets of transcripts (line 116 in file 178682_0_merged_1534147925.pdf), the authors should elaborate beyond the caption text for fig 1d including an exact definition of “high”, “medium” and “low”. If this is what is already described in Methods “RNA Secondary structure content and eCLIP”, then make an explicit pointer. Possibly substantiate.

We consider that this question was addressed by removing the grouping analyses (see replies above). As a result, **Figure 1d** does not longer exist (the new **Figure 1d** indeed reports a strong correlation between eCLIP and PARS structure).

- (11) For the conclusion drawn (lines 170-171 in file 178682_0_merged_1534147925.pdf) the authors need of substantiate how such a conclusion can be drawn from using 200 transcripts.

In the new analyses we found a trend at transcriptome level between transcripts structure (PARS) and BioGrid binders of the coded proteins (**Figure 3c**).

In lines 170 and 171 we previously indicated that proteins involved in several interactions are encoded by mRNAs that are bound by many proteins. During the revision of the manuscript we produced further analyses (see also **Point 5** of **Reviewer 2**) and our conclusion is now that there is regulatory level to be involved. **In essence, RNA binding to many proteins is highly regulated and, therefore, it is expected to be involved in many interactions.**

We addressed this subject in the Discussion section:

“Indeed, a tight relationship exists between the number of protein contacts of transcripts and the participation of encoded proteins in the network, which reveals an important level of transcriptional regulation³⁷ for highly connected genes⁴⁶: proteins involved in many interactions are encoded by mRNAs bound by several proteins. Our observation reveals an important functional connection, since highly contacted proteins participate in many cellular processes and require tight control at various post-transcriptional levels^{13,63,64}”

- (12) In lines 218-219 (file 178682_0_merged_1534147925.pdf) it is unclear precisely which RNAs is in which three classes and it would be helpful if the authors elaborate.

We have added a new class (background, the whole transcriptome) in the **Figure 3b** and extended the text to avoid confusion:

*“To understand if the correlation between protein-protein and protein-RNA interactions is a general property or simply a feature of the chaperone family, we analysed interactions of four RNA classes: (i) transcripts with no structure measured with PARS (“No structure” set); (ii) 100 transcripts with the highest structural values measured with PARS (“Top structure” set); (iii) the whole transcriptome (“background” set); and (iv) the set of 24 mRNAs coding for chaperone proteins (Genecards; <https://www.genecards.org>) for which we were able to obtain PARS data (“HSPs” set; **Methods, Fig. 3b**).”*

- (13) The reference #10 to multicleverMachine in the Methods document (Methods.pdf) is not the correct reference of the method. Please correct.

We corrected the reference accordingly.

(14) for all the methods mentioned in the main manuscript file (file 178682_0_merged_1534147925.pdf), e.g. multicleverMachine should be cited in there as well. The authors should sanitize if all methods mentioned are used.

We thank the Reviewer for raising this point. All the citations are now properly reported.

- *Minor:*

(15) Line 264: Perhaps “supports” instead of “demonstrates”.

Changed accordingly.

- *(16) For the programs used, e.g. catRAPID, CleverGO, multicleverMachine, etc, indicate precisely which version of the programs you used and indicate precisely all options given for how they were used in this manuscript. This should be indicated in the descriptions in the Methods section.*

Accordingly, we revised and updated the information about the programs employed (version, options, etc) in the Methods section.

References

1. Rouskin, S., Zubradt, M., Washietl, S., Kellis, M. & Weissman, J. S. Genome-wide probing of RNA structure reveals active unfolding of mRNA structures in vivo. *Nature* **505**, 701–705 (2014).
2. Rivas, E., Clements, J. & Eddy, S. R. A statistical test for conserved RNA structure shows lack of evidence for structure in lncRNAs. *Nat. Methods* **14**, 45–48 (2017).
3. Delli Ponti, R., Armaos, A., Marti, S. & Tartaglia, G. G. A Method for RNA Structure Prediction Shows Evidence for Structure in lncRNAs. *Front Mol Biosci* **5**, 111 (2018).
4. Seemann, S. E. *et al.* The identification and functional annotation of RNA structures conserved in vertebrates. *Genome Res.* **27**, 1371–1383 (2017).
5. Cid-Samper, F. *et al.* An Integrative Study of Protein-RNA Condensates Identifies Scaffolding RNAs and Reveals Players in Fragile X-Associated Tremor/Ataxia Syndrome. *Cell Rep* **25**, 3422-3434.e7 (2018).
6. Shabalina, S. A., Ogurtsov, A. Y. & Spiridonov, N. A. A periodic pattern of mRNA secondary structure created by the genetic code. *Nucleic Acids Res* **34**, 2428–2437 (2006).
7. Muppirala, U. K., Honavar, V. G. & Dobbs, D. Predicting RNA-protein interactions using only sequence information. *BMC Bioinformatics* **12**, 489 (2011).
8. Liu, Z. R., Wilkie, A. M., Clemens, M. J. & Smith, C. W. Detection of double-stranded RNA-protein interactions by methylene blue-mediated photo-crosslinking. *RNA* **2**, 611–621 (1996).
9. Van Nostrand, E. L. *et al.* Robust transcriptome-wide discovery of RNA-binding protein binding sites with enhanced CLIP (eCLIP). *Nature Methods* **13**, 508–514 (2016).
10. Marchese, D., de Groot, N. S., Lorenzo Gotor, N., Livi, C. M. & Tartaglia, G. G. Advances in the characterization of RNA-binding proteins. *Wiley Interdiscip Rev RNA* **7**, 793–810 (2016).
11. Mitchell, S. F. & Parker, R. Principles and properties of eukaryotic mRNPs. *Mol. Cell* **54**, 547–558 (2014).
12. Gerstberger, S., Hafner, M. & Tuschl, T. A census of human RNA-binding proteins. *Nature Reviews Genetics* **15**, 829–845 (2014).

13. Jeong, H., Mason, S. P., Barabási, A. L. & Oltvai, Z. N. Lethality and centrality in protein networks. *Nature* **411**, 41–42 (2001).
14. van der Lee, R. *et al.* Intrinsically disordered segments affect protein half-life in the cell and during evolution. *Cell Rep* **8**, 1832–1844 (2014).
15. Arraiano, C. M. *et al.* The critical role of RNA processing and degradation in the control of gene expression. *FEMS Microbiol. Rev.* **34**, 883–923 (2010).
16. Tartaglia, G. G., Pechmann, S., Dobson, C. M. & Vendruscolo, M. Life on the edge: a link between gene expression levels and aggregation rates of human proteins. *Trends Biochem Sci* **32**, 204–6 (2007).
17. Kato, M. *et al.* Cell-free formation of RNA granules: low complexity sequence domains form dynamic fibers within hydrogels. *Cell* **149**, 753–767 (2012).
18. Ouyang, Z., Snyder, M. P. & Chang, H. Y. SeqFold: Genome-scale reconstruction of RNA secondary structure integrating high-throughput sequencing data. *Genome Res.* **23**, 377–387 (2013).
19. Agostini, F. *et al.* catRAPID omics: a web server for large-scale prediction of protein-RNA interactions. *Bioinformatics* **29**, 2928–2930 (2013).
20. Cirillo, D., Agostini, F. & Tartaglia, G. G. Predictions of protein–RNA interactions. *WIREs Comput Mol Sci* **3**, 161–175 (2013).
21. Bellucci, M., Agostini, F., Masin, M. & Tartaglia, G. G. Predicting protein associations with long noncoding RNAs. *Nat. Methods* **8**, 444–445 (2011).
22. Lang, B., Armaos, A. & Tartaglia, G. G. RNAct: Protein–RNA interaction predictions for model organisms with supporting experimental data. *Nucleic Acids Res* doi:10.1093/nar/gky967

Reviewers' comments:

Reviewer #1 (Remarks to the Author):

The authors have addressed most of my concerns. I don't have any questions at the moment.

Reviewer #2 (Remarks to the Author):

The authors have done considerable additional work to improve the manuscript. My most pressing concerns are somewhat alleviated. I have a couple few minor comments.

1. Although many of my concerns are alleviated, it is still my opinion that the authors over "sell" the observed correlation between mRNA structure and RBP interactions without a clear biological implication, beyond just – more regulation -.
2. In my opinion, the condensate work isn't connected to the rest of the paper. This is neat result, but it doesn't really contribute to the main biological relevance as posited above.
3. The abstract no longer reflect the composition of the paper well (the experiments are highlighted, but now comprise a relatively small component of the work), all the additional work should be reflected in the abstract.
4. Is there some way to discriminate between the protein HSP70 and its transcript consistently in the text? I frequently forget that the authors use HSP70 to refer to the transcript (something about HSP being Heat Shock Protein) while reading this manuscript and have remind myself that when the authors say HSP70, they really mean the HSP70 transcript.
5. In my opinion, Figure 6 is not necessary.

Reviewer #3 (Remarks to the Author):

The authors have done substantial work in their revision, but regrettably not explicitly addressed my concern on using only the 100 least and most structured transcripts. The authors should when LS and HS sets are defined state why the 100 least and 100 most structured cases are needed (and shouldn't they replace HS with MS? Or replace "most structured" with "Highest Structured" ?)

Although the authors repeat the catRapid analysis with RPIseq on other non-similar proteins (please define what you mean by "non-similar"), the authors do still not provide a robustness analysis by repeating the analysis of least and most 150, 200, 300, ... structured transcripts. If the authors really did this then I assume that they would have explicitly mentioned that the repeat the catRapid analysis using e.g. the least and highest 300 transcripts. I don't think that the computational demands of catRapid justify to go beyond the 100 LS/HS. Supercomputing time is readily available at providers and could easily overcome this if it was a real obstacle increasing the number. If this was a real issue the authors could and in fact should state the computation time and the corresponding CPU architecture. This was not brought up as an issue in the original submission and neither in the revision.

Again on line 278, it seems as the analysis includes all transcripts with no structure , but in "(ii)" it is "only" the 100 highest. What happens if the authors use top 200, 300 etc.?

In the response to my concern (1) the authors state "All the points (a), (b), (c), (d), (f), (g) are fully addressed by the new analysis (see Figure 1d and Figure 1e)." I don't see that and I don't see why the authors did not respond one by one to this key concern. The same type of response is given for my concern (2). I do not clearly see this and wonder why the authors did not respond clearly item by item.

Other aspects which I find not clearly addressed include my previous comment that RNA can function through structured domains the authors reply that they normalize by the length, but the point is that its not necessarily the length and the fraction of transcripts these take up that is point, but merely whether the transcript has a structured domain regardless of the domain length, e.g. 5 domains each of length 30nt might be equally functional as 5 domains each of length 50nt, and if both are on a 1000 nt transcript the length normalization might not be appropriate to draw conclusions.

For estimating an FDR on the authors data, I don't see how that should conflict with the reviewers other comparisons (Fig 1b, 3f and 5b) as this could be made independently. I don't see how Fig 1b, 3f and 5b says something about the FDR in spite two groups are compared.

Reviewer #1 (Remarks to the Author):

- *The authors have addressed most of my concerns. I don't have any questions at the moment.*

We thank the Reviewer for his/her comments, especially those related to expanding the statistical analysis to a larger number of transcripts and using different experimental approaches for RNA secondary structure analysis.

Reviewer #2 (Remarks to the Author):

- *The authors have done considerable additional work to improve the manuscript. My most pressing concerns are somewhat alleviated. I have a couple few minor comments.*

Although many of my concerns are alleviated, it is still my opinion that the authors over “sell” the observed correlation between mRNA structure and RBP interactions without a clear biological implication, beyond just – more regulation –.

Many thanks for acknowledging the new additional work. We agree with the Reviewer that the biological implication of our findings should be further investigated. Yet, the central aim of our work was to show that RNA structure and number of protein interactions are intimately linked. In the current version of the manuscript we propose a series of biological implications: i) there is a direct link to mechanisms associated with gene regulation (**Figure 2b** and **Figure 3a** – ‘complex regulation’); ii) a tight relationship exists between the RNA type and the networks in which the protein product is involved (**Figures 3b** and **Figure 3d** – ‘chaperones’); iii) RNA secondary structure itself modulates the formation of ribonucleoprotein condensates (**Figure 4b** – ‘phase separation’).

While we agree with the Reviewer that it is difficult to report on all the biological implications of the correlation between mRNA structure and RBP interactions, we think that we opened the avenue for new studies by showing the most powerful application –interference with protein aggregation.

We have better highlighted the significance of our findings in the Introduction and Conclusions.

“At the transcriptome level, we found that the amount of RNA secondary structure correlates with the number of protein interactions. We propose several implications associated to this relationship: i) a link to RNA types and biological roles; ii) a connection to regulatory networks; and iii) the ability to modulate phase separation. Based on our observations, we also demonstrated that this RNA property can be exploited in vitro to tune the contact network of a protein aggregate.”

and

“The observations presented here, from transcriptome-wide to single molecule analyses, indicate that RNA controls gene regulation at multiple levels. The correlation between RNA structure and number of protein interactions could lead to the discovery of RNA functions that are presently unknown. As demonstrated in the case of protein aggregation, there are RNA-based mechanisms that control phase separation and could be important for the formation of membrane-less organelles. Overall, the complexity and diversity of protein-RNA networks reported here open the avenue for the investigation of regulatory processes.”

- *2. In my opinion, the condensate work isn’t connected to the rest of the paper. This is neat result, but it doesn’t really contribute to the main biological relevance as posited above.*

We thank the Reviewer for helping out in connecting the different parts of the paper, we have now improved the flow in the main text. Starting from our original observation that there is a relationship

between RNA structure and number of protein interactions, we formulated a series of hypotheses that we corroborated by computational analysis and led to the key experiment on how one structured RNA can modulate the formation of ribonucleoprotein condensates by directly binding to proteins.

We better highlighted the connection between hypothesis and practical application in the Introduction as well as different parts of the Results.

See previous reply and:

“Thus we hypothesize that a structured RNA, because of its higher protein-interacting potential, is able to affect the protein interaction network more than a poorly structured RNA. In a proof-of-concept experiment, we used a chemical compound, biotinylated isoxazole (b-isox) to induce formation of a liquid-to-solid phase transition of a protein assembly that we incubated with either HS (HSP70) or LS (BRaf) transcripts (Fig. 4a). We observed that HS altered the composition of the protein aggregate more than LS RNA (Fig. 4b and Supplementary Table 8).”

- 3. *The abstract no longer reflect the composition of the paper well (the experiments are highlighted, but now comprise a relatively small component of the work), all the additional work should be reflected in the abstract.*

Indeed, we have rewritten the abstract to better emphasize the flow.

“Here we used computational approaches to investigate the relationship between RNA structure and protein interactions. Using in vitro and in vivo experimental data, we show that the amount of double stranded regions in a transcript correlates with its ability of contacting proteins. This relationship, which we call structure-driven protein interactivity, allows to classify RNA types, identifies regulatory networks and may control the phase separation of ribonucleoprotein assemblies. We validate our hypothesis by showing that a highly structured RNA is able to rearrange the composition of a protein aggregate by binding to proteins in vitro.”

- 4. *Is there some way to discriminate between the protein HSP70 and its transcript consistently in the text? I frequently forget that the authors use HSP70 to refer to the transcript (something about HSP being Heat Shock Protein) while reading this manuscript and have remind myself that when the authors say HSP70, they really mean the HSP70 transcript.*

We have written *Hsp70* in italics to refer to the RNA and in regular font to mention the protein, using standard annotation for human genes.

- 5. *In my opinion, Figure 6 is not necessary*

We thank the Reviewer for raising this point. We think that it is important to graphically show and sum up our observations. However, we have realised that displayed alone may be out of context. For this reason, we included it in the **Figure 2** (panel e), where we present that many different biological levels are impacted by the relationship RNA structure and protein interactors.

Reviewer #3 (Remarks to the Author):

- *The authors have done substantial work in their revision, but regrettably not explicitly addressed my concern on using only the 100 least and most structured transcripts. The authors should when LS and HS sets are defined state why the 100 least and 100 most structured cases are needed (and shouldn't they replace HS with MS? Or replace "most structured" with "Highest Structured" ?)*

We are truly grateful for these remarks that helped to greatly improve the manuscript.

We have fully addressed the concern regarding the analysis of 100+100 RNAs. Indeed, the computational part (previously reported in Figure 1b and c and now in **Supplementary Figure 1**) served the only purpose of presenting a trend that was the investigated transcriptome-wide in **Figures 1c, 1d and 1e**. Following up the Reviewer's comment we modified **Figure 1b** to show the transcriptome-wide prediction of protein-RNA interactions. We revised the terminology when using HS and LS acronyms to indicate highest and lowest structural content, respectively

- *Although the authors repeat the catRapid analysis with RPIseq on other non-similar proteins (please define what you mean by "non-similar"), the authors do still not provide a robustness analysis by repeating the analysis of least and most 150, 200, 300, ... structured transcripts. If the authors really did this then I assume that they would have explicitly mentioned that they repeat the catRapid analysis using e.g. the least and highest 300 transcripts. I don't think that the computational demands of catRapid justify to go beyond the 100 LS/HS. Supercomputing time is readily available at providers and could easily overcome this if it was a real obstacle increasing the number. If this was a real issue the authors could and in fact should state the computation time and the corresponding CPU architecture. This was not brought up as an issue in the original submission and neither in the revision.*

We took up the challenge of showing that catRAPID can be predictive in large-scale analyses and for this purpose we used all the data available from a recent project (analogue of 120 years of calculations in a high-throughput cluster¹). The interaction predictions were calculated over several months on a shared set of 80 HP BL460c nodes with two Intel Xeon E5-2680 2.70 GHz CPUs and 120 GB of usable DDR3-1600 memory each, using 8 cores per cluster job. These are part of the CRG's high-performance computing cluster.

The results shown below (**Figure R1**) confirm the significance of the observed trends (p-values $\ll 0.001$; Wilcoxon test) for different thresholds, as requested by the Reviewer: 10% most-structured vs 10 least-structured RNAs, 15%, 20%, ..., 50% (full transcriptome).

Figure R1. *catRAPID* predictions¹ of protein interactions with all the human RNAs ranked by PARS structural content (HS=high structural content and LS= low structural content)² and for which eCLIP information is available³. The fractions 10%,15%, ... ,50% refer to the comparison between top HS vs bottom LS structural scores (p-values $\ll 0.001$; Wilcoxon test). The results indicate that *catRAPID* is able to distinguish HS and LS groups, in agreement with previous observations reported in **Figure 1b** and supporting the analysis of experimental data presented in **Figure 1e**.

The fraction, number HS+LS RNAs (equal size groups) and protein-RNA pairs is reported below:

10%	3289	101178
15%	4933	143112
20%	6577	184902
25%	8222	224682
30%	9866	263598
35%	11510	303435
40%	13155	341111
45%	14799	378714
50%	16444	414929

This remarkable result indicates that our approach is able to reproduce the trends reported in **Figures 1c, 1d** and **1e**. It is impossible to run these calculations for other methods such as *RPISeq*⁴, as they are only available in the form of webservers that allow calculations of a maximum of 100 interactions per run (for this purpose we prefer to use non-similar protein sequences, i.e. similarity scores <85% computed by CD-HIT⁵, we also indicate our definition of non-similar at the legend). Thus, previous comparisons between *catRAPID* and *RPISeq* will be moved to **Supplementary Figure 1**.

Although *catRAPID* performances are reported in a recent publication¹, we refer to previous papers for more details (and citations by others to them)⁴⁻⁷. We measured the False Discovery Rate FDR on all the interactions reported in **Figure R1** in comparison with eCLIP data (fdrtool <http://strimmerlab.org/software/fdrtool/index.html>)¹⁰. As shown in **Figure R2**, the FDR anti-correlates with *catRAPID* score, dropping below 0.1 when it lies between 2 and 3, which indicates strong predictive power. Importantly, not all protein-RNA interactions occur in the cell lines investigated by eCLIP, and it is possible that some false positives (i.e. predicted by *catRAPID* but not detected) are instead true positives (i.e. they would take place in different conditions or cells).

Figure R2. Comparison between *catRAPID* predictions¹ and interactions detected by eCLIP³. The false discovery rate drops below 0.1 when the *catRAPID* z-score lies between 2 and 3, thus indicating strong predictive power.

- Again on line 278, it seems as the analysis includes all transcripts with no structure, but in “(ii)” it is “only” the 100 highest. What happens if the authors use top 200, 300 etc.?

Accordingly, we divided the whole transcriptome in five consecutive regions based on the secondary structure content and showed a correlation with the number of protein interactions (**Figure R3**). The

analysis is reported in **Figure 3b** and indicates that RNAs coding for chaperones are highly structured and produce proteins with a large number of contacts. Previous Figure 3c is now showed as **Supplementary Figure 7**.

Figure R3. Relationship between secondary structure of human transcripts and interactions of the encoded proteins. RNAs coding for chaperones (HSPs) are highly structured produce proteins with a large number of contacts.

In addition, we also revised the terminology and removed the “no

structure” name replaced by Low Structured since the transcripts without structure were considered and not defined by PARS.

- *In the response to my concern (1) the authors state “All the points (a), (b), (c), (d), (f), (g) are fully addressed by the new analysis (see Figure 1d and Figure 1e).” I don’t see that and I don’t see why the authors did not respond one by one to this key concern. The same type of response is given for my concern (2). I do not clearly see this and wonder why the authors did not respond clearly item by item.*

As for the specific points made in the previous revision round:

- **[concern from previous round of revisions]** *For selected transcripts, the authors needs to:*

(a) justify that these transcripts non-redundant, e.g. neither close paralogues nor represent different isoforms in the same gene

With the analysis shown in **Figure R1** we provide the full transcriptome analysis and there is no need to justify the selection of particular isoforms present in the 100+100 RNA analysis.

- **[concern from previous round of revisions]** *(b) justify that these transcripts representable for the entire transcriptome*

In **Figure 1b** and **Figure 1c** we report the analysis for the full transcriptome so we do not need to provide a justification for a fraction of it.

- **[concern from previous round of revisions]** *(c) make sure that the more structured ones are not due to high GC content*

As anticipated in the previous round of revisions, there is a weak correlation between structure and GC content, which is fully justifiable considering the GC content is expected to be higher in stable sequences, as the interaction between G and C is the most energetically favourable (**Figure R4**)^{11,12}. This is indeed, what the Reviewer also suggests, as in another comment when he/she stated “As it is well known that a random sequence also fold in structure (e.g. more GC rich ones), the authors should take existing screens for evolutionary conserved RNA structures”.

When the analysis is applied to the whole transcriptome, the trend is still present (**Figure R4**). Thus, the result obtained by selecting the two extreme sets is just an indication of what we can observe in a more quantitative trend.

Figure R4. *We observe a weak relationship (correlation of 0.09) between GC and PARS structural contents, which is in agreement with previous reports indicating a contribution of GC pairing to RNA stability¹¹.*

- **[concern from previous round of revisions]** *(d) compare the average structured transcript, e.g. how do we know that the 100 least structured transcripts according to PARS are not outliers? In fact this could be one interpretation of the first peak of the bi-modal*

distribution in fig 1a. One can therefore with reason argue that it is the “tail” of the distribution with the right most peak the transcripts should be selected from in the range of approximately -13 and equal size area as the top end. The authors should repeat the analysis for this region as the LS set.

In **Figure 1b** the RNA structural content is compared with the number of protein interactions **transcriptome-wide**, thus there is no need to perform special investigations on a restricted pool of 100+100 RNAs. As the identified properties exist in a continuum range, there is no need to postulate the existence of outliers.

- **[concern from previous round of revisions]** (e) take the FDR of PARS in account and estimate if this (and possibly how) impact the conclusions.

Already discussed in our previous reply (**Figure R2**). Moreover, this point is not mentioned by the Referee in the new round of requests.

- **[concern from previous round of revisions]** (f) Consider if the sharp tale of the in the low end of fig 1a, could that represent transcripts that are strongly covered be proteins and therefore show up on PARS as unstructured?

This is truly an excellent point. Yet, since PARS experiments use native deproteinized RNA isolated from cells ², we can exclude this event.

- **[concern from previous round of revisions]** (g) Give in concordance with (b) a rationale for selecting just 100 sequences in each and in addition repeat the analysis for larger amount of selected transcripts in the LS and HS groups as in total 200 transcripts compared to e.g. GENCODEs ~59000 coding and non-coding genes and 206694 transcripts, rather small. The analysis should be repeated for group sizes of 250, 500, 1000, 2000 and 5000. The difference in structure preferences should still be seen although likely with less strength.

This has indeed been done accordingly, see **Figure R1**.

- The same type of response is given for my concern (2). I do not clearly see this and wonder why the authors did not respond clearly item by item.

[concern from previous round of revisions]. It remain unclear if the 100 transcripts selected in each end of the spectrum are equally represented by:

(a) non-coding and coding genes

We now provide an analysis for **all the transcriptome** in **Figure 1d, 1e** and **1f**. We did not operate any selection on the coding or non-coding fraction.

- **[concern from previous round of revisions]** (b) which of these groups share the same length (obviously longer sequences might have a bigger chance by being match by PARS data by random

We perform **transcriptome-wide** analysis:

We only found a very weak correlation of 0.10 between secondary structure and RNA length transcriptome-wide, (**Figure R5**). This is now reported in the main text:

“As for the PARS data, we found a weak correlation (< 0.10 ; Pearson’s) with RNA length and GC content, indicating that these two factors positively contribute to the secondary structure by increasing the size of the conformational space as well as the overall stability¹.”

Figure R5. We do not find correlation between structural content and transcript length.

(c) whether the coding sequences respectively have the same size UTRs both 5’ and 3’ independently, but also the total 5’+3’ UTR length per gene. If that makes up a bigger fraction of the gene it will be structured.

As previously stated regarding the UTRs, we found that removing the UTRs, the structural content does not change, as shown in

Figure R6 and R7 (Supplementary Figure 3).

Figure R6. RNA structural content with and without UTRs. We observe a correlation of 0.94 for all the transcriptome.

We address this in the text below:

“Although part of the mRNA structure is concentrated in the UTRs², when these are excluded, the distribution of the structural content does not change substantially (Supplementary Fig. 3; Pearson’s correlation between transcripts with and without UTRs = 0.94). The RNAs known to interact with proteins such as snRNAs³ and snoRNAs⁴ show high amount of structure, whereas RNAs targeting complementary regions in nucleic acids such as antisense, miRNAs and a number of long intergenic non-coding RNAs (lincRNAs)^{5,6} feature the smallest amounts of structure⁷ (Supplementary Table 4). In agreement with our findings, Seemann et al. previously observed a tight relationship between protein binding and conservation of structural elements in mRNAs, which occur to a lesser extent in long non-coding RNAs (lncRNAs)⁸.”

Figure R7. We do not find correlation between A) 3’ B) 5’ UTR length and overall structural content.

In agreement with this finding, we note that the experiments with *Hsp70* and *Braf* (Figure 4) RNAs were performed in absence of the UTRs. Removing the UTRs does not change the overall structural content (Figure 3c and Supplementary Figure 8).

Moreover, in the specific case of *Hbg1* and *Hbg2*, which was added during the last revisions (Figure 2b), we report:

“Interestingly, the increased double-stranded content in *HBG1*, especially at the 3’ UTR, is associated

with in an augment in interaction of translation regulatory elements (**Fig. 2b**) and a concomitant decrease in expression levels (NCBI Gene ID: 3048).”

- [concern from previous round of revisions] (d) These aspects should also be considered for the additional analysis in 1(f).

With reference to 1(f) “ Consider if the sharp tale of the in the low end of fig 1a, could that represent transcripts that are strongly covered by proteins and therefore show up on PARS as unstructured?”, we 1) do not use the 100+100 tails any more, but the whole transcriptome; 2) PARS experiments use native deproteinized RNA isolated from cells².

- Other aspects which I find not clearly addressed include my previous comment that RNA can function through structured domains the authors reply that they normalize by the length, but the point is that its not necessarily the length and the fraction of transcripts these take up that is point, but merely whether the transcript has a structured domain regardless of the domain length, e.g. 5 domains each of length 30nt might be equally functional as 5 domains each of length 50nt, and if both are on a 1000 nt transcript the length normalization might not be appropriate to draw conclusions.

We completely agree with the Reviewer. As in other theoretical works^{13,14}, we do not just normalize by length, but compute the fraction of double stranded regions over the entire sequence.

Indeed, PARS distinguishes double- and single-stranded regions using the catalytic activity of two enzymes, RNase V (able to cut double-stranded nucleotides) and S (able to cut single-stranded nucleotides)^{15,16}.

In formulas, given the stepwise function $\vartheta(x)=1$ for $x > 0$ and $\vartheta(x)= 0$ otherwise, we computed the fraction of structured domains as:

$$PARS \text{ structural content} = \frac{1}{L} \sum_i \vartheta \left(\log \frac{V(i)}{S(i)} \right)$$

We includes the formula and a comment to it in the methods of the manuscript.

- For estimating an FDR on the authors data, I don't see how that should conflict with the reviewers other comparisons (Fig 1b, 3f and 5b) as this could be made independently. I don't see how Fig 1b, 3f and 5b says something about the FDR in spite two groups are compared.

As now added to the main text “The FDR becomes highly significant for the most-stringent experimental set (FDR=0.1)” (**Figure 5b**). Yet, we do not think that additional analyses would be feasible or would add important value to the paper:

- While the False Discovery Rate FDR can be measured in cases such as those presented in **Figure R2** (around half a million of interactions), it is not meaningful for just 10-20 interactions as in **Figure 3f** and **Figure 5b**¹⁷.
- Not all protein-RNA interactions occur in the cell lines investigated by eCLIP, and it is possible that a number of false positives (i.e. predicted by *catRAPID* but not detected) are instead true positives (i.e. they would take place in different conditions).

1. Lang, B., Armaos, A. & Tartaglia, G. G. RNAct: Protein–RNA interaction predictions for model organisms with supporting experimental data. *Nucleic Acids Res* doi:10.1093/nar/gky967
2. Wan, Y. *et al.* Landscape and variation of RNA secondary structure across the human transcriptome. *Nature* **505**, 706–709 (2014).
3. Van Nostrand, E. L. *et al.* Robust transcriptome-wide discovery of RNA-binding protein binding sites with enhanced CLIP (eCLIP). *Nature Methods* **13**, 508–514 (2016).
4. Muppurala, U. K., Honavar, V. G. & Dobbs, D. Predicting RNA-protein interactions using only sequence information. *BMC Bioinformatics* **12**, 489 (2011).
5. Fu, L., Niu, B., Zhu, Z., Wu, S. & Li, W. CD-HIT: accelerated for clustering the next-generation sequencing data. *Bioinformatics* **28**, 3150–3152 (2012).
6. Cirillo, D. *et al.* Quantitative predictions of protein interactions with long noncoding RNAs. *Nat Meth* **14**, 5–6 (2017).
7. Bellucci, M., Agostini, F., Masin, M. & Tartaglia, G. G. Predicting protein associations with long noncoding RNAs. *Nat. Methods* **8**, 444–445 (2011).
8. Marchese, D., de Groot, N. S., Lorenzo Gotor, N., Livi, C. M. & Tartaglia, G. G. Advances in the characterization of RNA-binding proteins. *Wiley Interdiscip Rev RNA* **7**, 793–810 (2016).
9. Cirillo, D., Agostini, F. & Tartaglia, G. G. Predictions of protein–RNA interactions. *Wiley Interdisciplinary Reviews: Computational Molecular Science* **3**, 161–175 (2013).
10. Strimmer, K. A unified approach to false discovery rate estimation. *BMC Bioinformatics* **9**, 303 (2008).
11. Shabalina, S. A., Ogurtsov, A. Y. & Spiridonov, N. A. A periodic pattern of mRNA secondary structure created by the genetic code. *Nucleic Acids Res* **34**, 2428–2437 (2006).
12. Ouyang, Z., Snyder, M. P. & Chang, H. Y. SeqFold: Genome-scale reconstruction of RNA secondary structure integrating high-throughput sequencing data. *Genome Res.* **23**, 377–387 (2013).
13. Delli Ponti, R., Marti, S., Armaos, A. & Tartaglia, G. G. A high-throughput approach to profile RNA structure. *Nucleic Acids Res.* (2017). doi:10.1093/nar/gkw1094
14. Ponti, R. D., Armaos, A., Marti, S. & Tartaglia, G. G. A method for RNA structure prediction shows evidence for structure in lncRNAs. *bioRxiv* 284869 (2018). doi:10.1101/284869
15. Kertesz, M. *et al.* Genome-wide measurement of RNA secondary structure in yeast. *Nature* **467**, 103–107 (2010).
16. Spitale, R. C. *et al.* Structural imprints in vivo decode RNA regulatory mechanisms. *Nature* **519**, 486–490 (2015).
17. Benjamini, Y. & Hochberg, Y. Controlling the False Discovery Rate: A Practical and Powerful Approach to Multiple Testing. *Journal of the Royal Statistical Society. Series B (Methodological)* **57**, 289–300 (1995).

REVIEWERS' COMMENTS:

Reviewer #3 (Remarks to the Author):

The authors have made a thorough revision and addressed my requests and I'm happy to see the outcome.

I have a few minor things.

- 1) The fractions listed in the response letter (p.6 top) should be provided in a table in the supplementary material.
- 2) I requested that the authors indicate the compute resources spent, both CPU years and memory, while listing on which architecture it was carried out. This is listed in the response letter, but as far as I can see neither in the manuscript or in the supplementary material. I strongly encourage the authors to mention this explicitly in the main text, as it is critical that the readers are aware of the volume of calculations.
- 3) Line 360: plans  plants

REVIEWERS' COMMENTS:

Reviewer #3 (Remarks to the Author):

- *The authors have made a thorough revision and addressed my requests and I'm happy to see the outcome.
I have a few minor things.*

Thank you for the positive criticism that help us improve the manuscript.

- *1) The fractions listed in the response letter (p.6 top) should be provided in a table in the supplementary material.*

The corresponding table have been provided in **Supplementary Figure 1**.

- *2) I requested that the authors indicate the compute resources spent, both CPU years and memory, while listing on which architecture it was carried out. This is listed in the response letter, but as far as I can see neither in the manuscript or in the supplementary material. I strongly encourage the authors to mention this explicitly in the main text, as it is critical that the readers are aware of the volume of calculations.*

Now the compute resources indicated by the reviewer are explicitly mentioned in the Methods section.

- *3) Line 360: plans  plants*

The word has been corrected accordingly.